# Development of a sequential tool, LMDZ-NEMO-med-V1, to conduct global to regional past climate simulation for the Mediterranean basin: An Early Holocene case study

Tristan Vadsaria[1,3], Laurent Li[2], Gilles Ramstein[1] and Jean-Claude Dutay[1]

[1]Laboratoire des Sciences du Climat et de l'Environnement, CEA-CNRS- Université Paris Saclay, Gif-sur-Yvette, 91191, France
[2]Laboratoire de Météorologie Dynamique, CNRS-ENS-Ecole Polytechnique- Sorbonne Université, Paris, 75005, France
[3]Atmosphere and Ocean Research Institute, University of Tokyo, Kashiwanoha, Chiba, Japan

*Correspondence to*: Tristan Vadsaria (tristan.vadsaria@lsce.ipsl.fr)

## Abstract

Recently, major progress has been made in the simulation of the ocean dynamics of the Mediterranean using atmospheric and oceanic models with high spatial resolution. High resolution is essential to accurately capture the synoptic variability required to initiate intermediate and deep-water formation, the engine of the MTC (Mediterranean Thermohaline Circulation). In paleoclimate studies, one major problem with the simulation of regional climate changes is that boundary conditions are not available from observations or data reconstruction to drive high-resolution regional models. One consistent way to advance paleoclimate modelling is to use a comprehensive global to regional approach. However, this approach needs long-term integration to reach equilibrium (hundreds of years), implying enormous computational resources. To tackle this issue, a sequential architecture of a global-regional modelling platform has been developed for the first time and is described in detail in this paper. First of all, the platform is validated for the historical period. It is then used to investigate the climate and in particular, the oceanic circulation, during the Early Holocene. This period was characterised by a large reorganisation of the MTC that strongly affected oxygen supply to the intermediate and deep waters, which ultimately led to an anoxic crisis (called sapropel). Beyond the case study shown here, this platform may be applied to a large number of paleoclimate contexts from the Quaternary to the Pliocene, as long as regional tectonics remain mostly unchanged. For example, the climate responses of the Mediterranean basin during the last interglacial (LIG), the last glacial maximum (LGM) and the Late Pliocene, all present interesting scientific challenges which may be addressed using this numerical platform.

## 1 Framework of the study

### 1.1. Introduction

The Mediterranean basin is a key region for the global climate system. It  is considered to be a climate "hotspot" (Giorgi, 2006), due to its high sensitivity to global warming. In the past, it has been the seat

of important human civilisations, and it continues to play a very important role in international geopolitics with a dense population along its coasts. There is great diversity in the Mediterranean ecosystems, both marine and terrestrial. The Mediterranean region is also rich in paleoclimate records with a variety of proxies. Indeed, this area experienced major changes during the glacial-interglacial cycles (Jost et al., 2005; Ludwig et al., 2018; Ramstein et al., 2007). Another long-term cycle of changes due to high-frequency precession which drastically modified the hydrological patterns of this area (monsoon, sapropels) is also superimposed.

Due to the peculiarities of both the atmospheric and oceanic circulation in the region, high-quality climate modelling of the Mediterranean region needs to have high spatial resolution (Li et al., 2006). Indeed, the presence of strong gusts of wind in winter are essential to trigger oceanic convection and these can only be correctly represented in high-resolution models. Limited area models (LAM), or regional climate models (RCM), present some advantages in this regard, since they generally demand less computing resources, allowing them to be run at high spatial resolution for a given region. However, their usefulness for paleoclimate purposes is limited because of the lack of adequate lateral boundary conditions to drive the RCMs. The main reason why few comprehensive modelling exercises to explain paleoclimate changes around the Mediterranean have been performed is that the level of computing resources required for high resolution and long simulations is inaccessible. This is especially true in the case of the Mediterranean Thermohaline Circulation (MTC), which has significantly changed in the past, at both centennial and millennial scales.

Here we describe a modelling suite to define high-resolution atmospheric conditions over the Mediterranean basin from global ESM (Earth System Model) paleoclimate simulations. This atmospheric forcing can then be used to run a highly resolved ocean model (NEMOMED8 $1/8°$) to accurately simulate ocean dynamics. This tool allows us to achieve a high spatial resolution and equilibrated simulations with a run time of 100 years. The objective of this study is to develop a modelling platform sufficiently comprehensive to conduct paleoclimate studies of the Mediterranean basin. The potential of this platform is illustrated by investigating climate situations from the present period and from the Early Holocene that is supposed to generate sapropel events.

The sapropel events provide excellent case studies on the impact of global changes on the Mediterranean basin. These periodic events are related to a long period of anoxia of the deep and bottom waters triggered by an enhancement of the African monsoon caused by periodicities of the orbital precession. However, the localisation of the forcing source caused by orbital variability is still a subject of debate. This is especially true for the last sapropel, denoted S1, which occurred during the early Holocene (between 10500 and 6800 ka BP) (De Lange et al., 2008). Reproducing past climate variations over the

Mediterranean basin, including the sapropel events, is therefore a challenge for the modelling
community.

The paper is organised as follows: In the first section, we briefly review the different approaches used
to simulate the Mediterranean climate and sea conditions, and we also present the concept of the
sequential procedure that we propose. Section 2 presents in detail the model architecture we developed.
Finally, we present applications with simulations of the historical period (1970-1999) in Section 3 and
the Early Holocene (around 9.5 ka) in Section 4.
**1.2. Overview of current Mediterranean Sea modelling**
The Mediterranean Sea, due to its limited size and its semi-enclosed configuration, has a faster
equilibrium response ($10^2$ years) than the global ocean ($10^3$ years). Because of this semi-enclosed
configuration, there are a few requirements that modelling of the Mediterranean Sea needs to satisfy so
that its evolution can be properly represented. High resolution in both the atmospheric forcing and the
oceanic configuration is necessary to correctly simulate the convection areas and the associated
thermohaline circulation (Lebeaupin Brossier et al., 2011; Li et al., 2006). Depending on the mechanism
studied, the resolution of the ocean model used by the research community ranges from ¼° (e.g. for
paleo-climatic simulation), to 1/75° (for hourly description of the mixed layer, tide-based investigation).
The results for oceanic convection are highly dependent on the flux of heat, flux of water, and the wind
stress at the air-sea interface especially the seasonal variability and intensity. There are many modelling
configurations in the scientific literature making it impossible to provide an exhaustive review of all of
them. We can summarise them by presenting the different approaches used to drive the Mediterranean
oceanic model, along with their advantages and drawbacks. We underline our new, coherent method,
which captures the changes in ocean dynamics in the Mediterranean basin derived from global
paleoclimate simulations.

*Observations and reanalysis*
The most common way to simulate the general circulation of the Mediterranean Sea is to run a regional
oceanic general circulation model forced by surface fluxes and wind stresses derived from observations
and reanalyses. In this way, an oceanic model can be driven by realistic fluxes. In most cases, this implies
an observation-based reconstruction of relevant variables with a spatial atmospheric resolution of less
than 50 km and a daily temporal resolution, at a minimum, in order to simulate the formation of dense
water (Artale, 2002). This approach is adapted to simulate the present-day Mediterranean Sea and to
explore the complexity of its sub-basin circulation and water mass formation (Millot and Taupier-
Letage, 2005). However, it is not well adapted to the study of past and future climate, partly due to the
excessive computing resources needed.
*Atmospheric model*
A second method consists of forcing a regional oceanic model with simulations from an atmospheric
model, AGCM (Atmospheric Global Climate Model) or ARCM (Atmospheric Regional Climate
Model). Since the AGCM resolution (typically 100 to 300 km horizontally) is coarse, statistical and/or
dynamical downscaling is usually needed, especially for wind-stress so that the ORCM (Ocean Regional
Circulation Model) can be correctly forced (Béranger et al., 2010). Currently, dynamical downscaling
with ARCM is the preferred option because it generally improves simulations of the climate in the
Mediterranean region and especially of the hydrological cycle (Li et al., 2012).
This configuration is broadly used to assess anthropogenic climate changes (Adloff et al., 2015; Macias
et al., 2015; Somot et al., 2006). In these studies, the Mediterranean Sea simulations are generally driven
by the outputs of an ARCM, which is, in turn, driven by the GCM or observation-based reanalysis. It
should be noted that biases in oceanic variables can be reduced through constant flux correction (Somot
et al., 2006). This configuration is suitable for high-resolution simulation of the past Mediterranean Sea
(Mikolajewicz, 2011 for the LGM; Adloff et al., 2011 for the Early Holocene among others).
*Regional coupled model*
Although the majority of the Mediterranean Sea models are ocean-alone models, some of them use a
coupled configuration between the Mediterranean Sea and the atmosphere. Such a coupled configuration
generally improves the simulation of the air-sea fluxes, including their annual cycle (de Zolt et al., 2003),
but may show climate drifts in key parameters such as the SST. Regional coupled models are now
emerging as a tool in Mediterranean climate modelling (Artale et al., 2010; Dell'Aquila et al., 2012;
Drobinski et al., 2012; Sevault et al., 2014; Somot et al., 2008). However, this full-coupling
configuration is currently not possible for high-resolution paleoclimate issues requiring long simulation
for hundreds or thousands of years.
*Importance of boundary conditions*
The boundary conditions applied to the Mediterranean Sea domain, in particular, the exchanges of water,
salt and heat with the Atlantic Ocean through the Strait of Gibraltar modulate significantly the
Mediterranean circulation (Adloff et al., 2015). This is especially true at the millennial scale where
deglaciation episodes and fluctuations of the AMOC (Atlantic Meridional Overturning Circulation) and
the Mediterranean Sea affect each other (Swingedouw et al., 2019). The level of discharge from the
main rivers is also crucial as is illustrated by the sapropel episodes, where an increase in freshwater
input drastically slowed down the MTC. Most of current models impose prescribed (observed when
possible) conditions in the near Atlantic zone, including temperature and salinity. The same
methodology can be used to prescribe river discharges. However, it must be acknowledged that
determining inputs from rivers into the Mediterranean Sea, either of water or other materials, still
presents serious challenges for modelling.
**1.3. Concepts for a sequential procedure to perform global-to-regional modelling**
In this paper, a new architecture for high-resolution modelling of the climate of the Mediterranean basin
for past, present and future conditions is proposed. This architecture is based on a method that provides
as much compatibility as possible amongst the models used and high consistency with data.

*Step 1: Global climate*
Our goal is to simulate different climate conditions for the Mediterranean basin. The first step of any
relevant procedure should be to simulate the global climate conditions from which the simulation of the
regional climate is driven. These may be already available in simulations from previous PMIP exercises
for various periods (e.g. mid-Holocene, Last Glacial Maximum, Last Interglacial and mid-Pliocene) as
well as for different sapropel events and interglacials (e.g. MIS11, MIS13 and MIS19). However, this
is not always possible due to the large volume of high-frequency 3-D atmospheric circulation variables
involved. An alternative approach, used in some regional climate simulations (Chen et al., 2011;
Goubanova and Li, 2007; Krinner et al., 2014), consists of using an AGCM (either an independent one
or the same one used for the global climate simulation) run with appropriate values for global Sea
Surface Temperature (SST) and Sea Ice cover (SIC), derived from PMIP global simulations. SST is
crucial to determine atmospheric features and responses, while SIC plays a key role in determining the
global albedo. Monthly SST and SIC are necessary and sufficient to drive an AGCM. They can be
acquired from global climate simulations or through a bias-correction procedure.

*Step 2: Regional climate*
After running an AGCM, regional climate can be now reproduced with an ARCM nested into the high-
frequency outputs from the AGCM. Of course, the ARCM can be run in parallel to the AGCM, or with
a small time delay. Thus, we avoid a large accumulation of intermediate information between the AGCM
and the ARCM. In our study, we assume that there would be no feedback from the regional scale to the
global scale, so only a "one-way" transfer of information (from global to regional) is considered. In our
case, the ARCM is a strongly zoomed-in version of the AGCM and is also driven by monthly SST and
SIC values, as used for AGCM. The higher resolution of the ARCM allows the synoptic variability and
seasonality of the Mediterranean region to be depicted so that a realistic wind pattern and hydrological
cycle may be reproduced. This approach provides a general framework for use in many different
paleoclimate periods from the Pliocene to the Pleistocene, as long as the basin tectonics remain
unchanged.

*Step 3: Mediterranean Sea Circulation*
Daily air-sea fluxes and wind stress provided by the ARCM are used as surface boundary conditions to
drive the ORCM to investigate the oceanic dynamics of the Mediterranean. It is reasonable to assume
that the boundary conditions of these air-sea fluxes represent the long-term trends of the oceanic
dynamics. Rivers may be considered interactive or not depending on the investigative objectives: runoff
can be prescribed from climatology or obtained from the hydrological component of the surface model.
Again, we highlight that our architecture does not include any feedback, between either the regional
ocean and the regional atmosphere, or the regional ocean and the global ocean. This configuration means
that we can avoid dealing with certain issues, for example, the influence of the Mediterranean Outflow
Water on the North Atlantic Ocean but is well adapted to provide consistent river runoff associated with
changes in continental precipitation.
**2       Model architecture**
An ensemble of modelling tools that includes two atmospheric models and a regional oceanic model is
used. Figure 1 summarises the configuration and shows the experimental flowchart.
**2.1. The atmospheric models (AGCM and ARCM)**
LMDZ4 (Hourdin et al., 2006;  Li, 1999) is the atmospheric general circulation model developed and
maintained by IPSL (Institut Pierre Simon Laplace). It has been widely used in previous phases of CMIP
and PMIP projects. The resolution of the model is variable. Its global version used here (referred to as
LMDZ4-global) is 3.75° in longitude and 2.5° in latitude with 19 layers in the vertical. It provides the
boundary conditions to drive LMDZ4-regional. LMDZ4-regional (Li et al., 2012) is a regionally-
oriented version of LMDZ4 with the same physics and same vertical discretisation, dedicated to the
Mediterranean region. The zoomed-in model covers an effective domain of 13°W to 43°E and 24°N to
56°N with a horizontal resolution of about 30 km inside the zoom. The rest of the globe outside this
domain is considered to be the buffer-zone for LMDZ4-regional where a relaxation operation is
performed to nudge the model with variables from the AGCM, at a 2-hour frequency. The resolution of
LMDZ4-regional decreases rapidly outside its effective domain. In both LMDZ4-global and LMDZ4-
regional, land-surface processes, including the hydrological cycle, are taken into account through a full
coupling with the surface model, ORCHIDEE (Krinner et al., 2005).
**2.2 The regional oceanic model (ORCM)**
NEMOMED8 (Beuvier et al., 2010; Herrmann et al., 2010) is the regional Mediterranean configuration
of the NEMO oceanic modelling platform (Madec, 2008). The horizontal domain includes the
Mediterranean Sea and the nearby Atlantic Ocean which serves as a buffer zone (from 11°W to 7.5°W).
The horizontal resolution is 1/8° in longitude and 1/8°cosφ in latitude, i.e. 9km to 12km from the north
to the south. The model has 43 layers of inhomogeneous thickness (from 7 m at the surface to 200 m in
the depths) in the vertical. River discharges are accounted for as freshwater fluxes in the grids
corresponding to the river mouths.  A dataset of climatological river discharges was proposed by default
to cover the entire Mediterranean draining basin with 33 river mouths. It is of course switched off when
rivers are interactive in the platform. The interactive calculation of freshwater discharges from rivers by
the land-surface model, ORCHIDEE, includes 192 river mouths that cover the Mediterranean draining
basin. The Black Sea, not included in NEMOMED8, counts as a river dumping freshwater into the
Aegean. The deposit rate is calculated based on total runoff into the Black Sea, plus the net budget of
precipitation (P) minus evaporation (E) over the Black Sea.

When the oceanic model NEMO is used alone, with prescribed surface fluxes, it is indispensable to
implement a restoring term with a constant coefficient of 40 W.m$^{-2}$.K$^{-1}$(as defined in Barnier et al. 1995)
. This is a standard procedure for NEMO to prevent eventual run-away cases. In our modelling chain,
the target temperature for the restoration is the surface air temperature from the regional atmospheric
model LMDZ4-regional.

227        **2.3 Modelling Sequence**

As shown in Fig. 1, the first step in our modelling chain is to obtain SST and SIC values from an Earth
System Model simulation able to reproduce global climate (for the past, present or future). We can
reasonably hypothesise that major global climate information can transit from global SST and SIC. This
hypothesis was deemed legitimate for climate downscaling purposes for Antarctic and Africa, in Krinner
et al. (2014) and Hernández-Díaz et al. (2017) respectively. In the present work we use IPSL-CM5A
(Dufresne et al., 2013) to extract relevant SST and SIC values to drive the AGCM (LMDZ4-global) and
the ARCM (LMDZ4-regional). The next step is to run the two atmospheric models, LMDZ4-global and
LMDZ4-regional, in the usual way as proposed by the AMIP community. This is the most expensive
step, as atmospheric models are the most demanding in terms of computing resources. Fortunately, it is
not necessary to run them for a long time as the atmosphere reaches equilibrium quickly. We applied 30
years of  simulation to both models. We consider this duration to be long enough to depict climate
variability for the simulation of past events. The AGCM nudges the ARCM in the conventional way of
one-way nesting for temperature, humidity, meridional and zonal wind every two hours. The nudging is
done using an exponential relaxation procedure with a timescale of half an hour outside the zoom and
10 days inside the zoom. Table S2 in the SOM summarises the forcings used, especially the orbital
forcing and atmospheric CO$_2$.
The necessary variables (surface air temperature, wind stress, P-E over the sea, heat fluxes) are provided
by ARCM to NEMOMED8 (ORCM) at daily frequency. The salinity and temperature conditions are
provided in three dimensions in the Atlantic buffer zone, near the Gibraltar Strait, and updated every
month. River runoff, updated every month, depends on the configuration used (prescribed climatological
rivers, or interactive rivers). Table S3 in SOM details these boundary conditions.
It is worthy to mention the work of Mikolajewicz (2011) who used a similar modelling chain (from a
coarse-resolution earth system model to a high-resolution regional oceanic model) to simulate the
Mediterranean Sea climate during the last glacial maximum. However, Mikolajewicz (2011) used only
an AGCM (ECHAM5) as the intermediate step. In our case, we found that the use of ARCM was
indispensable to produce high-quality forcing to correctly simulate the oceanic convection in
NEMOMED8.
**2.4    Bias correction**
The sequential modelling chain, despite the lack of interactivity and feedback at interfaces, allows for
error removal and bias correction at each step of the methodology. This adjustment is sometimes crucial,
especially when model outputs need to be of very high quality to be incorporated into impact studies.
This concept was further described in Krinner et al. (2019), as illustrated in Fig. 16 of their paper.
Therefore, to enhance our confidence in the realism of the simulation results, bias-correction may be
introduced when necessary. The correction method used in the present work generally follows the
conventional procedure, which is based on the difference between the model outputs for present day
simulations and actual observations. Biases corrected in this way, theoretically only valid for the
historical simulation (named HIST hereafter), are assumed to remain unchanged for past and future
simulation scenarios. However, the transferability between past and future periods is questionable. There
is no guarantee that the model error for one period is the same for other periods, even though the model
physics may be the same. In addition, paleodata are often rare and incomplete, and so, are unsuitable for
evaluation and correction of model errors. The most reliable basis is that established for the present day.
The reader can find a full description of the bias corrections and their eventual use in our applications
in the supplementary online material, "Text S2: Bias correction".


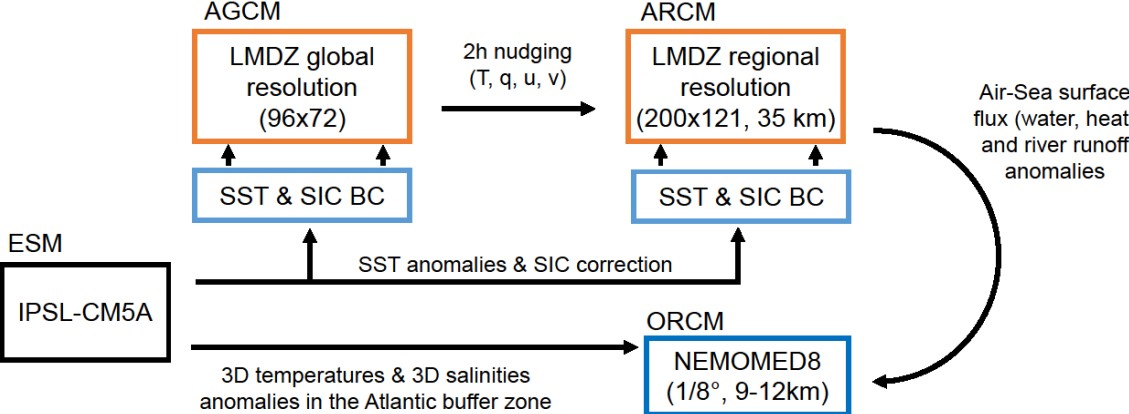



**Figure 1: Flowchart of the modelling chain including the four main components generally represented by ESM, AGCM, ARCM and ORCM, respectively, and actually implemented in our platform by IPSL-CM5A, LMDZ-global, LMDZ-regional and NEMOMED8. BC: boundary condition, u: zonal wind, v: meridional wind, q: specific humidity, T: temperature, S: salinity, SST: sea surface temperature, SIC: sea-ice concentration.**

## 3 Validation of the modelling chain for present-day climate 1970-1999

In this section, the capacity of the model to reproduce the climate of the recent past is evaluated, in particular, its ability to simulate sea surface characteristics as well as the Mixed Layer Depth (MLD) and oceanic convection patterns as these are key elements to reproduce the evolution of the Mediterranean Sea in past climate conditions.

### 3. 1 Experimental design

For the HIST experiment, SST and SIC observations (ERA-Interim, Dee et al., 2011) are used to force the AGCM. River runoff is from the climatology of Ludwig et al., (2009). Monthly mean climatological sea temperatures and salinities (World Ocean Atlas database from Locarnini et al., 2013, Zweng et al., 2013) are used for the Atlantic boundary zone. HIST atmospheric simulations for both global and regional simulations have a duration of 30 years. The length of the HIST oceanic simulation is also 30 years, but obtained after a 150-year spin-up. The forcings for each experiment are detailed in "Tables S2 and S3" in the supplementary online material. Spin-up phases for each simulation are also shown from "Figure S4" to "Figure S8" for the overturning stream function and the index of stratification.

## 3.2    Evolution of temperatures

Figure 2 depicts the temporal evolution, between 1970 and 1999, of annual mean surface air temperatures at two metres in the atmospheric simulations (global and regional) compared to observations for the whole globe and over the Mediterranean region. The two models reproduce a range of temperatures similar to the observations, with the Mediterranean temperatures warmer than the global temperatures. The global simulation (continued red curve in Fig. 2), after SST bias correction, is very close to the observation (continued black curve), with a tremendous improvement compared to IPSLCM5A (green curve in Figure 2) The regional model reproduces the warming trend and aspects of the interannual variability close to observations, but with a mean cold shift of about -0.6°C.

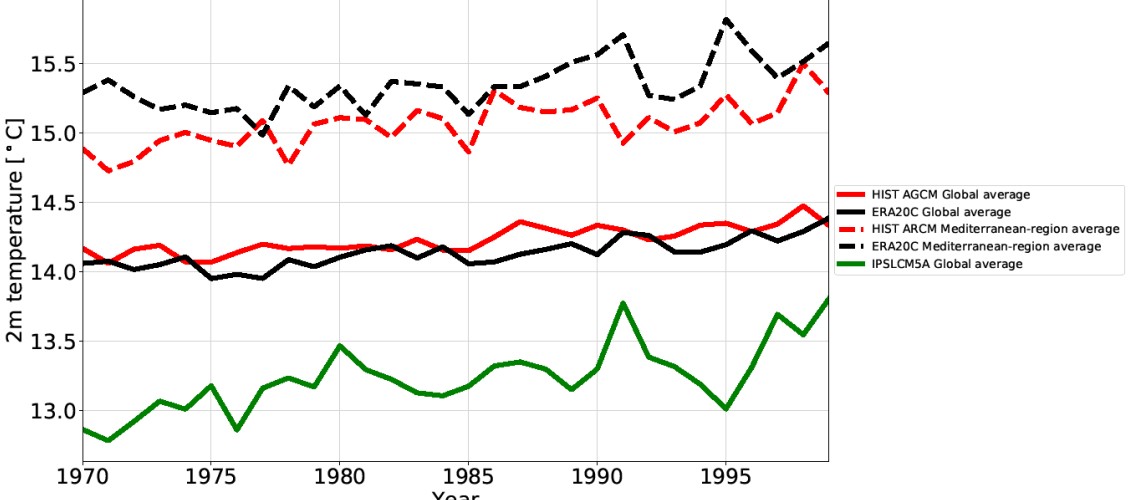

**Figure 2: Time series of annual mean surface air temperatures at 2 m in HIST (red) and ERA20C (black, ref: Stickler et al., 2014) and IPSLCM5A (green) for global average (solid lines) and Mediterranean-region (10°W- 35°E, 20°N-50°N) average (dashed lines).**

**3.3 Precipitation and freshwater budget**

Figure 3 a and b show the average annual precipitation for 1970-1999 in HIST over the Mediterranean region and the differences with observations. The main features of the distribution of precipitation over the Mediterranean region are simulated, in particular the distinct contrast between the very low precipitation in the southern region and higher precipitation in the north. The ARCM tends to generate higher precipitation than the AGCM due to the resolution refinement. Compared to observation, AGCM is closer to ERA20C (Stickler et al., 2014), whereas ARCM is closer to GPCP data (Adler et al., 2018). However, the regional model still overestimates the amount of precipitation, especially at 42°N, from 45° to 50° N, at 8°E and 20°E. It corresponds to most of Europe, especially over the Alps, the Pyrenees,

the Balkans and other mountainous regions. The freshwater budget over the Mediterranean Sea from
observations (a synthesis from Sanchez-Gomez et al., 2011 and from other sources) and in the various
simulations conducted in this study are summed up in Table 1. The simulated continental precipitation
is overestimated, but both the precipitation and evaporation over the Mediterranean Sea in HIST are
very close to the observations. The two other simulations included in Table 1, PICTRL and EHOL, are
those designed to investigates the Early Holocene climate (see Section 4).

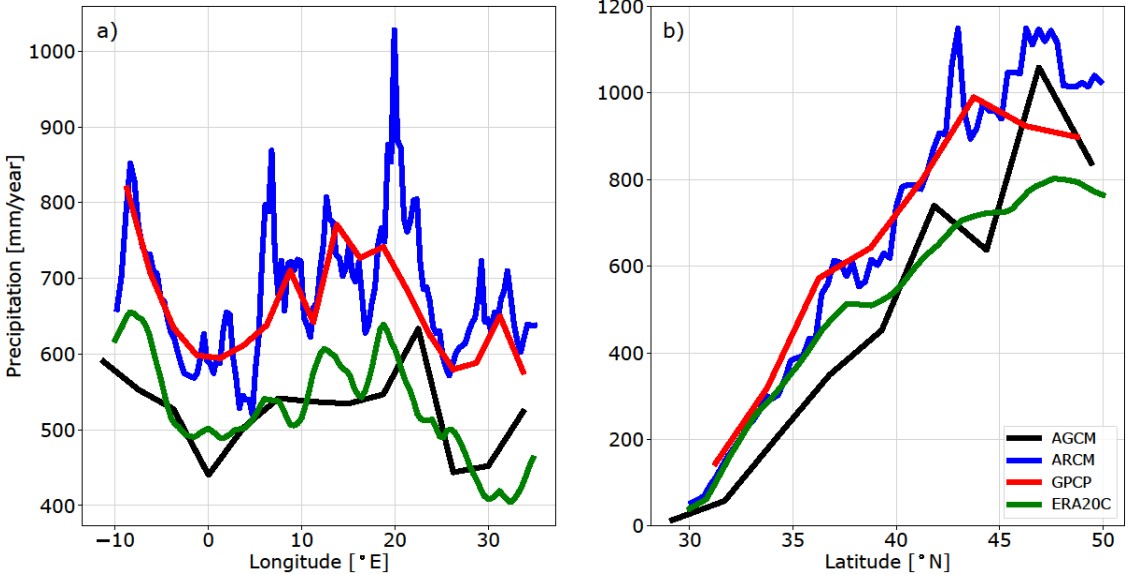


**Figure 3:  Annual mean precipitation, a) meridionally averaged (30 to 50°N), b) zonally averaged (-10 to 35°E), in the historical simulations with AGCM (LMDZ-global) and ARCM (LMDZ-regional). Observation comes from GPCP (Global Precipitation Climatology Project, 1979 to 1999, blue line, ref: Adler et al., 2018). and ERA20C (green line, ref: Stickler et al., 2014).**


| Dataset or experiment | E | P | R | B | E – P – R - B |
|---|---|---|---|---|---|
| OBS | 1096-1136 | 256-595 | 102-142 | 73-121 | 238-705 |
| HIST | 1106 | 443 | 74 | 104 | 485 |
| PICTRL | 1031 | 451 | 98 | 104 | 378 |
| EHOL | 1094 | 460 | 225 | 104 | 305 |

**Table 1: The Mediterranean Sea freshwater budget, expressed as mm.year⁻¹ for the whole water area (about 2.5 million of km²). E, evaporation, P, precipitation, R, river runoff, B, Black Sea discharge into the Mediterranean Sea. OBS is a summary from Sanchez-Gomez et al., (2011) for**

**P, E and P-E, from Ludwig et al., (2009) for R, from Lacombe and Tchernia, (1972), Stanev et al.,**
**(2000) and Kourafalou and Barbopoulos, (2003) for B. River discharges in HIST are from the**
**climatology of Ludwig et al., (2009). PICTRL uses the Nile of its pre-industrial (pre-damming)**
**value, 2930 m³.s⁻¹, annually (Rivdis database, Vorosmarty et al., 1998). River discharges in EHOL**
**are deduced from the difference between EHOL and PICTRL.**

### 3.4 Mediterranean Sea surface characteristics

Figure 4 displays the temperatures and salinities of the Mediterranean Sea simulated in HIST and the
deviations from observations. The model is able to capture the main characteristics of the pronounced
west-east gradient of SSS in the Mediterranean Sea (Figure 4 a). Values are within the range of
observations (mean bias = 0.32 PSU, error = 0.37 PSU, table 2). In the simulation, the Aegean Sea is
not salty enough (about -1.5 PSU) and the Adriatic/Ionian Sea is too salty (+1 PSU).
The model reproduced the northwest to southeast temperature gradient, as shown in Figure 4b. However,
the model shows a general cold bias (from -0.5 to -1.5 °C) over the entire Mediterranean (Figure 4e),
due to the cold bias already observed for the air temperature at 2m in the regional atmospheric forcing
(cf Figure 2).


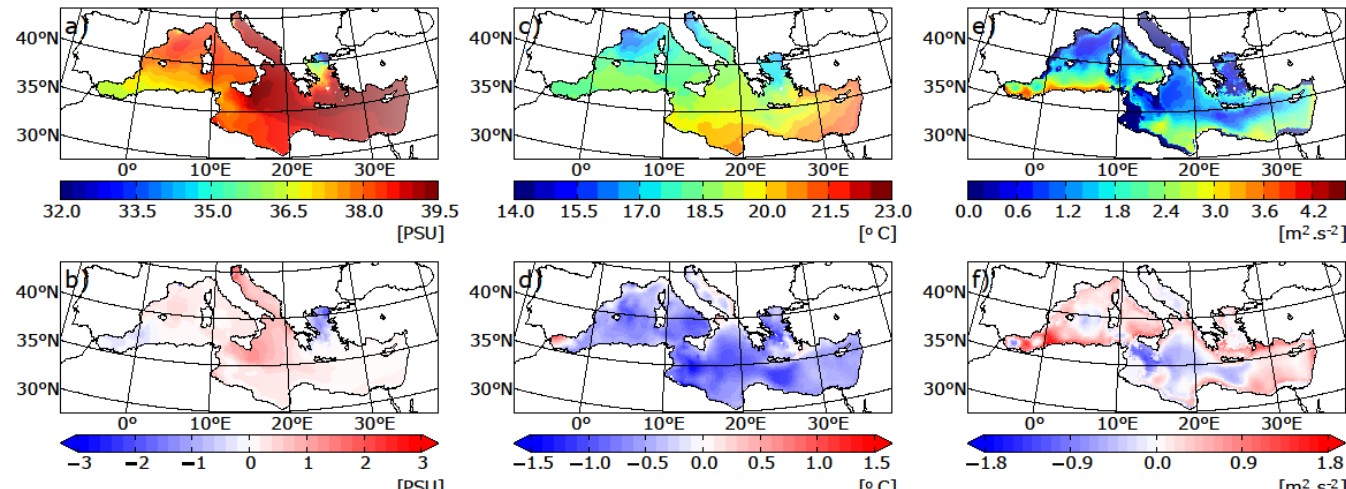


**Figure 4: Annual mean sea-surface salinity (left panels, SSS in PSU), sea-surface temperature**
**(middle panels, SST in °C) and index of water column stratification (right panels, winter IS in**
**m².s⁻²) simulated in HIST (top panels) and the HIST deviation (model – obs) from the observation-**
**based MEDATLAS data (averaged over the entire simulation).**


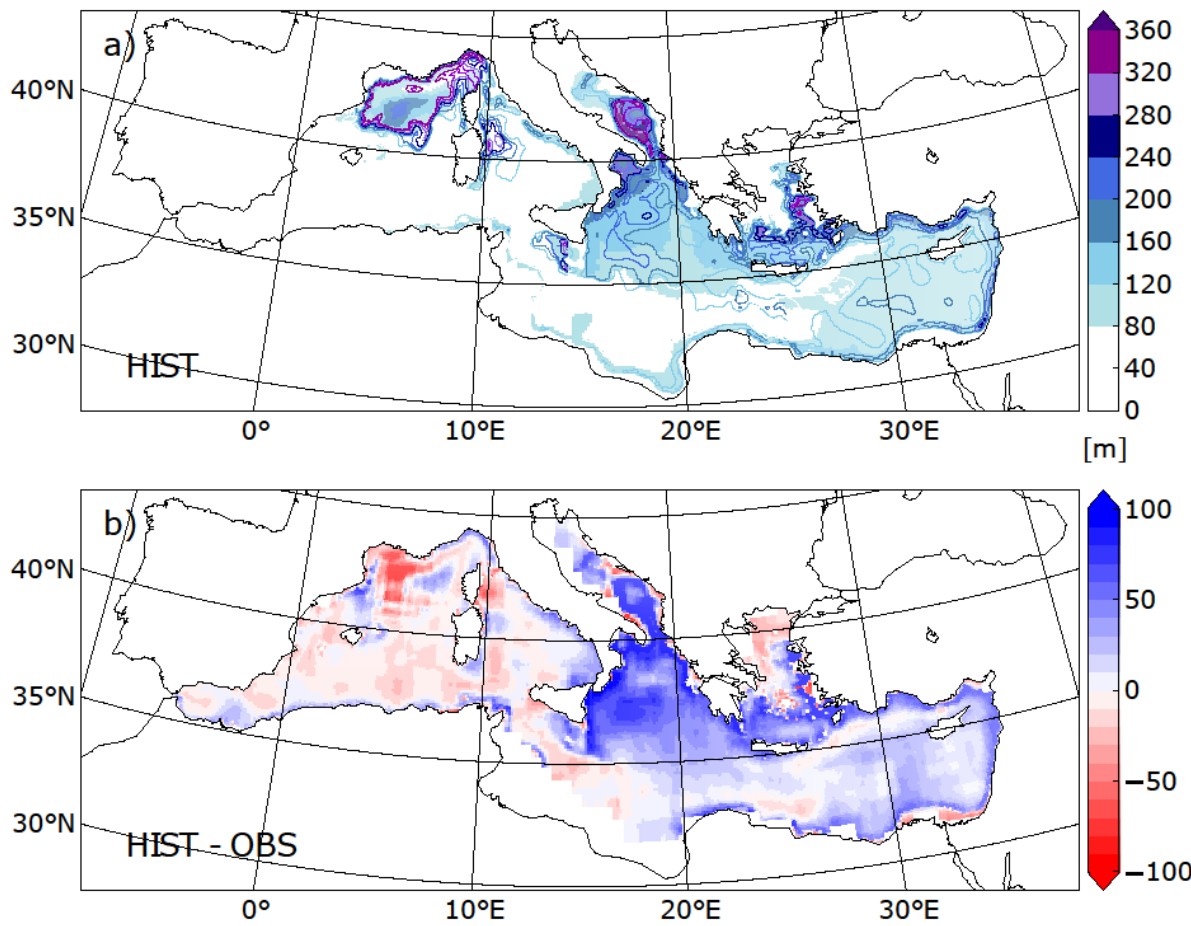

**Figure 5: a) Mixed layer depth simulated in HIST (panel a, in m) and as deviation (b) of HIST from observations of Houpert et al., (2015) averaged over the entire simulation for JFM (January February March). Contour lines in the upper panel a) represents the maximum of MLD throughout the HIST simulation.**

|  | SST (°C) | SSS (PSU) | IS (m².s⁻²) |
|---|---|---|---|
| **Mean bias (model – obs)** | -0.64 | 0.32 | 0.91 |
| **RMS error** | 0.45 | 0.37 | 0.29 |

**Table 2: Mean biases of sea surface temperature (SST), sea surface salinity (SSS) and index of stratification (IS) in the HIST simulation, expressed as the deviation from observations (MEDATLAS-II), and associated root mean square errors.**

### 3.4    Mediterranean Thermohaline circulation

Here, the general characteristics of the simulated thermohaline circulation is evaluated in regions where deep and intermediate water formation occurs. Figure 4c displays the stratification index (IS[1]) for HIST. IS is a vertical integration of the Brunt-Vaisala frequency. A lower IS implies that convection is more likely. The range of IS biases (Figure 4f), is from -1 to 1 $m^2.s^{-2}$ (mean bias = 0.91 $m^2.s^{-2}$, error = 0.29 $m^2.s^{-2}$). The model satisfactorily reproduces the convection in known intermediate and deep-water formation areas, namely the Gulf of Lions, the Adriatic Sea, the Ionian Sea, the Aegean Sea and the North Levantine.

Comparison with observations of the mixed-layer depth (Houpert et al., 2015) confirms that the model reproduces realistic intermediate and deep-water formation patterns, with a thicker MLD in the eastern basin, due to salty condition (Figure 4a and e), and a shallower MLD in the Gulf of Lions (figure 5b).

The simulated Mediterranean overturning circulation is analysed (figure 6). The Zonal Overturning stream Function (ZOF[2]) in figure 6a depicts the surface and intermediate circulation and the intermediate/deep circulation. The surface current from the Strait of Gibraltar flows up to 30°E and back to the Atlantic Ocean in the intermediate layers, through the Levantine Intermediate Water (LIW) outflow. Figure 6 c, e, and g represents the Meridional Overturning stream Function (MOF[3]) in the Gulf of Lions, the Adriatic Sea and the Aegean Sea, respectively. The surface cell in the longitude-depth plan is comparable to previous studies done with the same regional oceanic model, but with different forcings (Adloff et al., 2015; Somot et al., 2006): the mean strength of the surface cell ranges from 0.8 to 1.0 Sv, and the longitudinal extension is from 5°W to 30°E. The simulated intermediate and deep cells are recognized in existing studies as having different characteristics. Our simulated pattern is very close to a similar historical run in Adloff et al., (2015), but is weaker than a historical run in Somot et al., (2006), and a second historical configuration (with refined air-sea flux) in Adloff et al., (2015). The ZOF in

---

[1] $IS(x, y, h) = \int_0^h N^2(x, y) z \, dz$. $N^2$ is the Brunt-Väisälä frequency. IS is calculated at each model grid $(x, y)$ for a given depth $h$ (set as the bottom of the sea, or as 1000 m when the depth is greater than 1000 m).

[2] $ZOF(x, z) = \int_h^z \int_{ys}^{yn} u(x, y, z) \, dy \, dz$. u is the zonal currents, h is the depth of the bottom, yn and ys are the north and south coordinates respectively.

[3] $MOF(y, z) = \int_h^z \int_{xe}^{xw} v(x, y, z) \, dx \, dz$. v is the meridional currents, h is the depth of the bottom, xw and xe are the west and east coordinates respectively.

HIST depicted in figure 6) is consistent with the reanalyses (1987-2013) of Pinardi et al. (2019) over
the Western basin, but shows a weaker Eastern deep cell compared to the reconstruction.

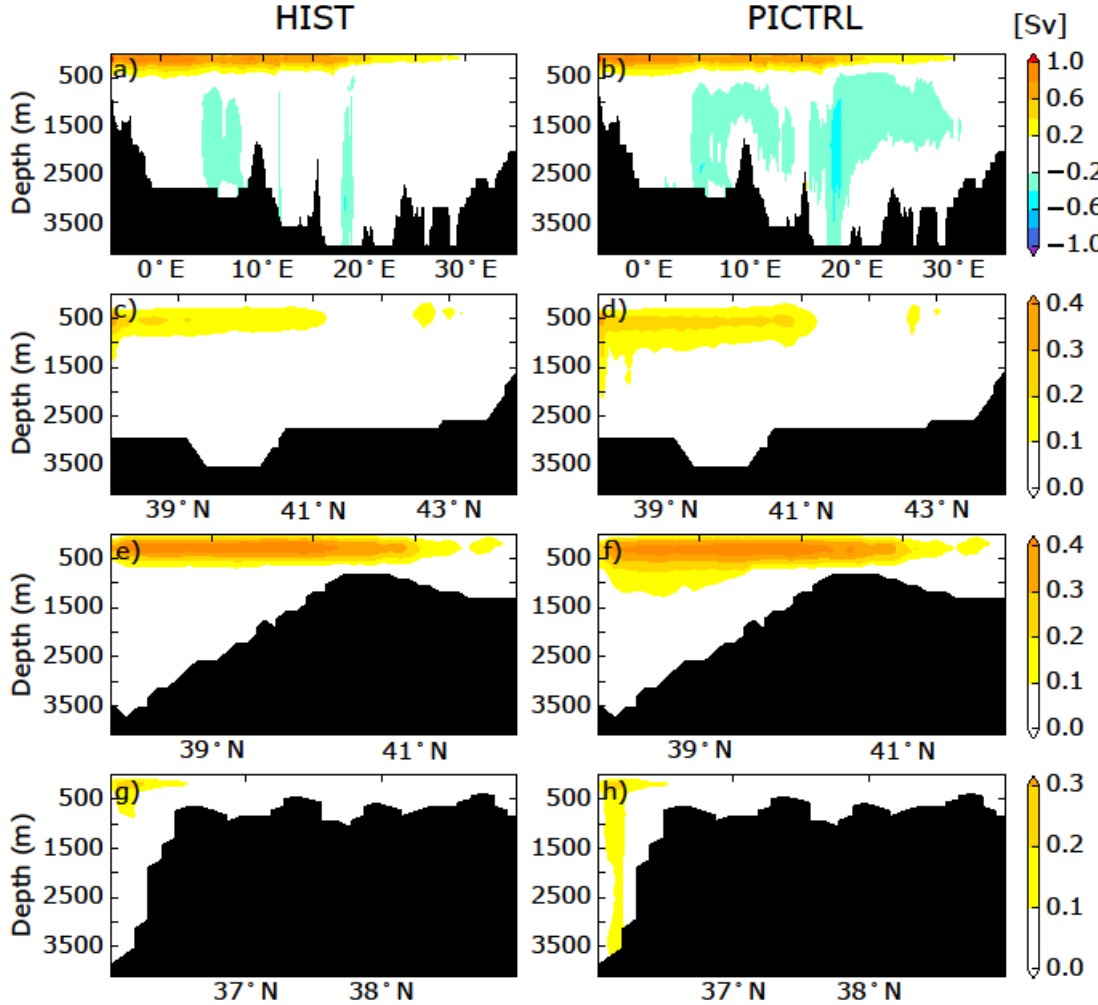


**Figure 6: a, b, Zonal Overturning stream-Function (ZOF) integrated from north to south and**
**shown as a longitude-depth section for the whole Mediterranean Sea, for HIST, and PICTRL**
**simulations (from top to bottom), respectively. Other panels show Meridional Overturning**
**stream-Function (MOF) shown as a latitude-depth section, integrated west/east for the Gulf of**
**Lion (c and d, longitudinal extent: 4.5° to 8°E), the Adriatic/Ionian Sea (e and f, 12° to 21°E), and**
**the Aegean Sea (g and h, 24° to 28°E) averaged over the entire simulation for HIST and over the**
**last 30 years of simulation for PICTRL.**

### 3.5 Summary of Validation

Validation of our platform was based on the historical period, 1970 to 1999. The atmospheric simulation is acceptable compared with observations for the air temperature at 2m at both global and regional scales. The simulated precipitation from the atmospheric models produces a signal that has the same range of variability as the observations, but there is significant overestimation of precipitation over the mountainous area and over the land surrounding the Mediterranean Sea. However, the freshwater budget over the sea is close to observations for both evaporation and precipitation. The areas of intermediate and deep convection produced by the model are realistic, and the simulation of the thermohaline circulation is well captured by the oceanic model and in the range of the state-of-the-art existing Mediterranean regional models (compared to the simulations of Adloff et al., 2015 and Somot et al., 2006 for instance) and reanalysis as well (Pinardi et al., 2019). These features inspire confidence in our modelling platform for the investigations of past climate.

## 4  Application of the modelling chain to the Early Holocene

In this section, results obtained when our sequential modelling chain is applied in a paleoclimate context are presented, which was our initial motivation for developing this modelling tool. We chose to test the performance of our tool on the Early Holocene, a period marked by significant changes in climate and ocean dynamics over the Mediterranean basin, when the last sapropel event, S1, occurred in the Mediterranean Sea. Our experimental design relies on the comparison of two simulations: the Early Holocene (EHOL) with PICTRL based on pre-industrial conditions, the latter acting as a reference.

### 4.1    Experimental design

As indicated in the general flowchart of our modelling platform, global SST and SIC are required to initiate our sequential modelling. The basic assumption is that the climate change signal can be reconstructed from global SST and SIC, an accepted practice within the climate modelling community. In this study, two existing long-term coupled simulations from IPSL-CM5A is used, one covering the pre-industrial period and the other covering the Early Holocene (around 9.5 ka). Taking the last 100 years of each simulation, a climatological SST and SIC is constructed. After conducting bias-correction, these outputs from IPSL-CM5A are then used to drive the AGCM (LMDZ-global) and the ARCM (LMDZ-regional) in a further step. The duration of the PICTRL and EHOL atmospheric simulations is 30 years (both global and regional models).

Oceanic temperature and salinity in the Atlantic buffer-zone, as well as freshwater discharges from Mediterranean rivers, are all bias-corrected for NEMOMED8, as described in the general methodology. However, it needs to be pointed out that the reference point for the Nile river discharge is not modern

observations but is set at pre-industrial values (2930 $m^3.s^{-1}$ for annual mean, Vorosmarty et al., 1998)
corresponding to a period before construction of the Aswan dam. The oceanic simulation is 90 years for
EHOL and 30 years for PICTRL, performed after a 200-year spin-up of PICTRL.
**4.2    Climate features depicted in LMDZ-global (AGCM)**
Because Early Holocene simulations are mainly driven by insolation forcing, an important feature is the
model response to seasonal temperatures. Figure 7 shows the difference between EHOL and PICTRL,
as reproduced in the AGCM, LMDZ-global, for the summer/winter temperature, JJAS precipitation and
JAS surface runoff. The atmospheric model imprints a stronger seasonality due to the increased Early
Holocene summer insolation. Warmer summer temperatures over Europe and North Africa (+ 6 °C,
figure 7b) and lower winter temperatures over Africa (-2 °C, figure 7a) reflect this feature. Variations
of the precession also trigger an enhancement of the African Monsoon (+ 10 $mm.day^{-1}$ over the Ethiopian
region, figure 7c). The main consequence of this increase in precipitation is an enhanced surface runoff
over the Ethiopian region. This hydrological state is similar to the African Humid Period caused by the
enhanced African Monsoon and the resultant increase in surface runoff, as shown in Rossignol-Strick et
al. (1982).

Our results are similar to those of previous modelling exercises for the Early- and Mid-Holocene (e.g.
Adloff et al., 2011; Bosmans et al., 2012; Braconnot et al., 2007; Marzin and Braconnot, 2009). They
are also consistent with various reconstructions of mid-Holocene precipitation (Harrison et al., 2014).
A detailed comparison can be made with the Early Holocene simulation reported in Marzin and
Braconnot (2009) which used for their experiment the same orbital parameters and the same atmospheric
model as EHOL. However, their model was coupled to an oceanic model, while an atmospheric model
and prescribed SST and SIC as boundary conditions are used in this study. Generally speaking, our
results for both surface air temperature and precipitation are very similar to those of Marzin and
Braconnot (2009), attesting to the validity of our approach using a simple atmospheric model
constrained by boundary conditions.  In the ensemble of PMIP simulations, available for the Early
Holocene and mid-Holocene, there are some robust outputs for the climate response to orbital forcing
but there are also some weaknesses common to most of the models (Braconnot et al., 2007; Kageyama
et al., 2013). One of these weaknesses is the underestimation of the spread of the African monsoon
towards North Africa. However, the increased discharge from the Nile river, induced by the enhanced
monsoon is well supported by data (Adamson et al., 1980; Revel et al., 2014; Williams, 2000).

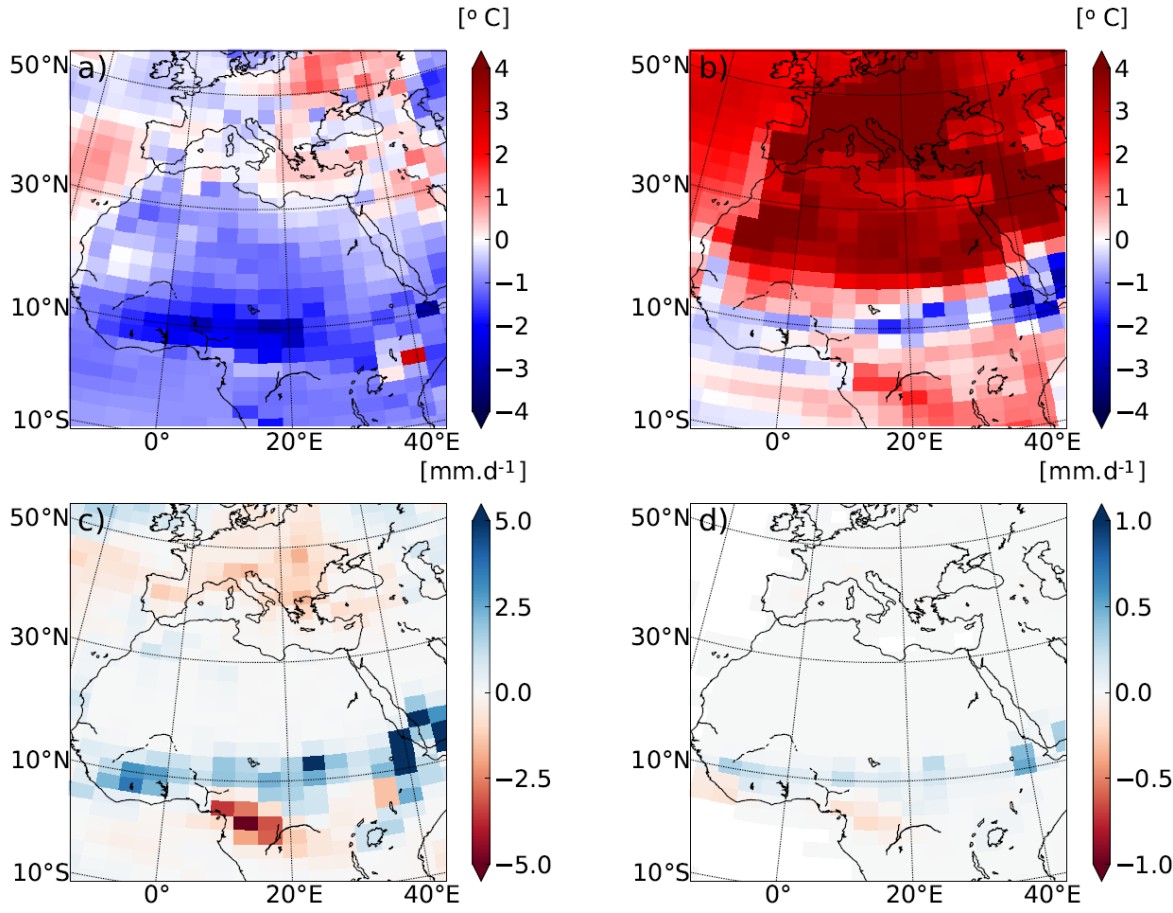

**Figure 7: Temperature and precipitation deviations of EHOL from PICTRL in LMDZ-global, the AGCM for a) winter surface air temperatures at 2 m, b) summer surface air temperatures at 2 m, c) June to August precipitation, and d) July to September surface runoff (averaged over the entire simulation).**

## 4.3    Mediterranean climate features with dynamical downscaling refinement

Figures 8, 9 and 10 show the results from the regional atmospheric model (LMDZ-regional), compared to those from LMDZ-global for PICTRL and EHOL over the Mediterranean region. In both the global and regional simulations, an increased seasonality is depicted, with warmer summer (+2 to +6 °C) and colder winter, especially over land (-3 to -1 °C, Figure 8). Downscaling with LMDZ-regional slightly reduces the amplitude of the summer warming and shows a more homogenous signal in winter over land. The general circulation of the surface wind in PICTRL is west to east (Figure 9b), in line with the dominant winter regime of westerlies in the region. This important feature is almost missed in the global model (Figure 9a) which reproduces a lower intensity than the regional model. The winter precipitation in EHOL, for ARCM (LMDZ-regional), increases over land in the Balkans and Italy and over the Adriatic, Ionian and Aegean Seas (Figure 10b). These changes are also present in the AGCM (LMDZ-

global) that, furthermore, shows an increase in Spain and Portugal (Figure 10a). It is in summer that the two models show the largest differences. In ARCM (LMDZ-regional), the Mediterranean basin experiences drier conditions, except in Italy and the North of the Balkans. Over the sea, precipitations slightly increase in EHOL (Figure 10). However, the AGCM (LMDZ-global) shows drier conditions in the northern two thirds of the Mediterranean domain, with more humid conditions in the southern third (Figure 10c). Changes in precipitation lead to unavoidable modifications in the runoff and river discharge into the Mediterranean Sea.

Although it is not straightforward to compare our "snapshot" simulations against environmental records (often used to reconstruct a timeline), our results compare well with the available data for this area (see supplementary online material, "Text S3: Comparison of model simulation outputs and reconstructed data for the Mediterranean basin"). Numerous proxies provide information on lake levels, paleo fires, pollen, isotopic signals recovered from speleothems which together describe the Mediterranean climate in the past. All of these proxies need to be brought together to provide a clear impression of the Mediterranean climate for this period (Magny et al., 2013; Peyron et al., 2011). Magny et al. (2007), based on records from Lake Acessa (Italy), suggested that aridification took place around 9200–7700 cal BP. Zanchetta et al. (2007), based on data recovered from speleothems in Italy, conclude that the Western Mediterranean basin experienced enhanced rainfall during the S1 (10000-7000 cal BP). Jalut et al. (2009), using pollen data, suggest that the summers were short and dry and that there was abundant rainfall in winter (autumn and spring as well) and remarked that these wetter conditions favoured broad-leaf tree vegetation. Different proxies seem to provide contradictory information and therefore, seasonality must be introduced to reconcile them. Peyron et al., (2011) mentioned wet winters and dry summers during the 'Holocene optimum'. Magny et al., (2013) support the hypothesis of seasonal contrast based on the analysis of multi-proxies.

Our EHOL simulation successfully depicts this temperature contrast between winter and summer. Precipitation is enhanced in winter. In summer, the Mediterranean region is globally drier, except over Northern Italy and the northern Balkans. As explained above, there is no precipitation signal over Northern Africa, although evidence of paleo-lakes has been found in Algeria (Callot and Fontugne, 1992; Petit-Maire et al., 1991), Tunisia (Fontes and Gasse, 1991) and Libya (Gaven et al., 1981; Lézine and Casanova, 1991) during the Early Holocene indicating increased rainfall in this area. In the supplementary material, a comparison between simulated continental precipitation outputs and pollen reconstruction data is provided. This comparison shows that the winter precipitation anomalies are consistent in both cases but that there is a distinct difference in summer values due to the more contrasted summer in the EHOL simulation.



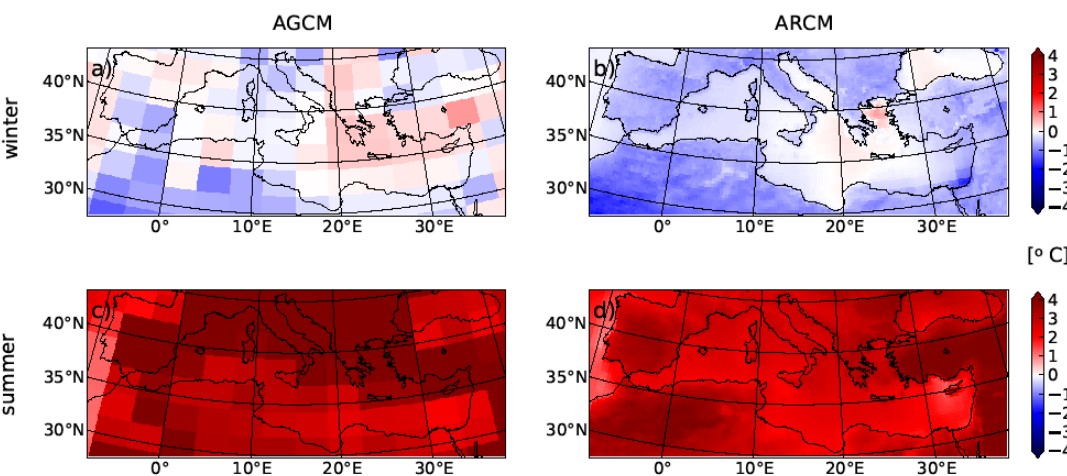


**Figure 8: Deviations (EHOL − PICTRL, averaged over the entire simulation) of surface air temperature at 2 m for winter (upper panels) and summer (lower panels), respectively. AGCM (LMDZ-global) is displayed on the left and ARCM (LMDZ-regional) on the right.**



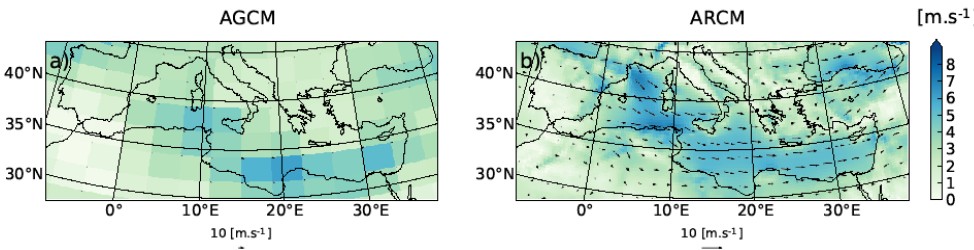


**Figure 9: Winter wind-speed in PICTRL for a) the AGCM and b) the ARCM.**


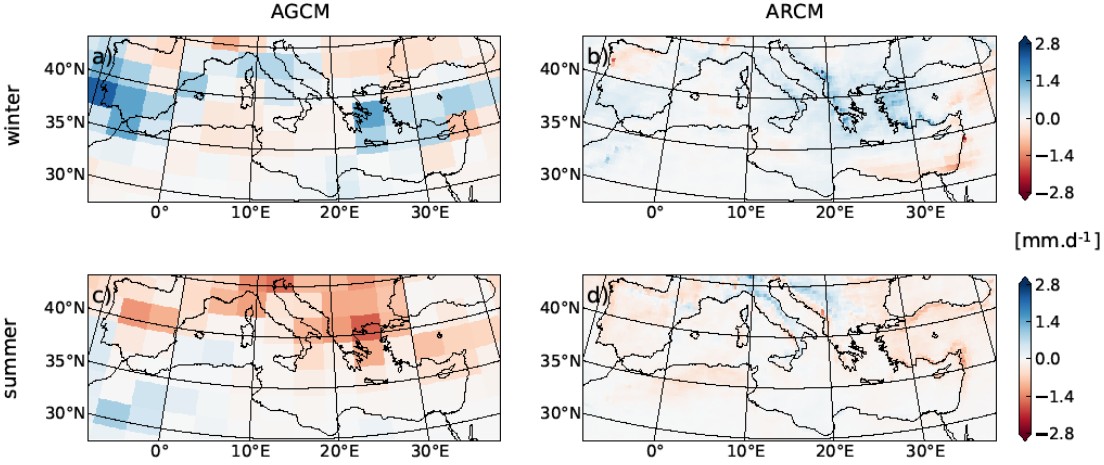

**Figure 10: Same as in Figure 8, but for precipitation rate (mm/day).**

### 4.4 Hydrological changes

Figure 11 shows anomalies (EHOL – PICTRL) of river freshwater supplies into the Mediterranean basin as simulated by the ARCM (LMDZ-regional). Bars are displayed for each calendar month to show the strong seasonal variation, and for the western and eastern basins separately. Due to their particular role and their specific treatment in our current modelling practice, the Nile and the Black Sea are also shown for the eastern basin, but not accounted in the sum. The North African rivers are not displayed since they don't show much changes for their catchment area. The Nile River shows important seasonal variation, with increase in summer and autumn and decrease in winter and spring. The Albanian rivers (Drini, Mat, Dures, Shkumbin and Vjosa) as well as the Vardar and the Buyukmenderes, produce positive anomalies in EHOL in winter, due to enhanced winter land precipitation in this simulation (Figure 10 b and d). The Black Sea net freshwater supply also changes in EHOL with important decreases in January, February, March and July, but increase in April. In EHOL, the supplementary winter freshwater input is less pronounced for the western basin than for the eastern basin (Figure 11b), but major rivers, such as Rhone and Ebro, do show a strong seasonal cycle. As a whole the western basin sees an increase of river discharges from March to June.

In terms of areal means for the entire Mediterranean draining basin, the different components of the freshwater budget are shown in Table 1 (bottom) for both PICTRL and EHOL, to be compared to the observation-based estimation OBS and the historical simulation HIST. From PICTRL to EHOL, the annual precipitation over the Mediterranean Sea itself does not change much, but the annual evaporation amount shows a slight increase (from 1031 to 1094 mm.year$^{-1}$). However, the most remarkable feature

is the increase of river discharges: 98 mm.year$^{-1}$ in PICTRL to 225 mm.year$^{-1}$ in EHOL. The total water
deficit finally decreases from 378 to 305 mm.year$^{-1}$.




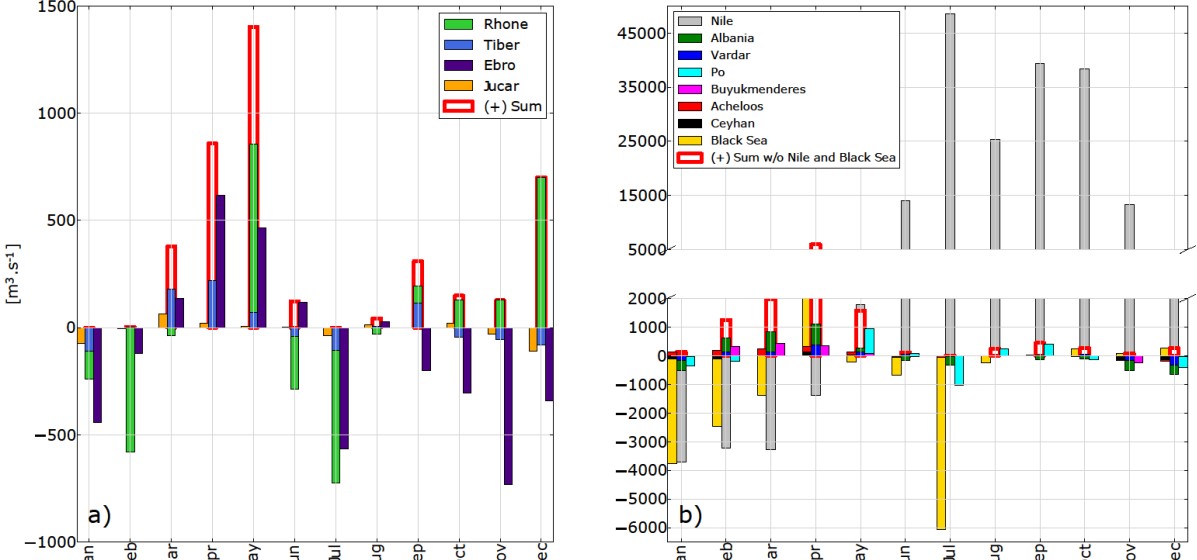


**Figure 11:    Monthly anomalies (EHOL – PICTRL, with seasonal variation) of fresh water**
**discharges (m$^3$.s$^{-1}$) for major rivers flowing into the western basin (left panel) and the eastern**
**basin (right panel). The sum of all rivers for each basin is also plotted. The Nile and the Black Sea**
**are also shown as rivers of the eastern basin, but not accounted into the basin-scale sum.**


**4.5    Changes in water properties of the Mediterranean Sea**
At the end of our modelling chain, changes in the properties of the Mediterranean seawater produced by
NEMOMED8 for PICTRL and EHOL are examined. It is important to mention at this stage, that for the
correction of the river runoff the reference is the pre-industrial state, and not the historical simulation
(as is the case for SST and SIC). Our aim was to keep river runoff anomalies free of anthropogenic
influence. In addition, the fact that the "pre-industrial" Nile river runoff (in other words before
damming) is well known influenced this choice. Figure 12 shows changes (EHOL minus PICTRL) for
sea surface salinities, index of stratification and MLD for the last 30 years of simulation. The EHOL
simulation reasonably reaches the steady state in terms of IS, ZOF and SSS, as shown in Figures S6 to
S8 of the supplementary material. The freshwater inputs from the Nile and the north-eastern margin
imply a lower salinity in the eastern basin. This decrease in salinity enhances stratification throughout
the Mediterranean Sea (with the exception of the Sicily Sea) and affects the convection areas by
decreasing the MLD, especially in the Gulf of Lions, in the Adriatic and Ionian Seas and in the Aegean.
Such a situation is expected and consistent with the basic climatology of MLD, shown in Figure 5. This
global stratification in EHOL is followed by a general reduction in the thermohaline circulation
compared to PICTRL (ZOF and MOF, Figure 13).

Numerous studies have documented the sapropel event S1 and the state of the Mediterranean Sea that
caused it. Emeis et al. (2000) mentioned a decreased SSS during this period in both the eastern and
western basins (as did Kallel et al., 1997 in the Tyrrhenian basin). In the subsection "*Sea Surface*
*Temperatures*" and "*Sea Surface Salinity*" of the section "Text S3"in the supplementary online material,
simulated SST and SSS to reconstructions are compared. Although simulated SST is in good agreement
with the reconstructed data, there is a gap between the simulated SSS and reconstructions. This
discrepancy is not surprising. Indeed, there are many explanations for the underestimation in our model
of the salinity. One of them is a common weakness in Early to Mid-Holocene simulations, namely, the
underestimation of the northward spread of the African monsoon and therefore, the underestimation of
the freshwater flow from North Africa. Adloff (2011), already pointed to a shortfall in freshwater input
to reconcile the simulated and observed SSS during the Early Holocene. Our oceanic simulation depicts
these behaviours well and is overall similar to previous modelling studies with lower resolution (Adloff
et al., 2011; Bosmans et al., 2015; Myers et al., 1998).

Two other issues need to be discussed for the Early Holocene. The first one is sea level, which was 20
metres lower than the present day  (Peltier et al., 2015). For the sake of simplicity, this difference of sea
level is not taken into account in the EHOL simulation. The second issue is the timing of the
(re)connection between the Black Sea and the Aegean Sea. This topic is still being debated. Sperling et
al. (2003) suggested this reconnection occurred around 8.4 ka BP, while by the calculations of Soulet et
al. (2011) it happened around 9 ka BP. Other studies found that an overflow from the Black Sea likely
occurred before this reconnection due to Fennoscandian ice-sheet melting during the deglaciation
(Chepalyga, 2007; Major et al., 2002; Soulet et al., 2011). For the purposes of this work, the Bosphorus
is maintained open in EHOL simulation, with the water exchange set at its modern value.




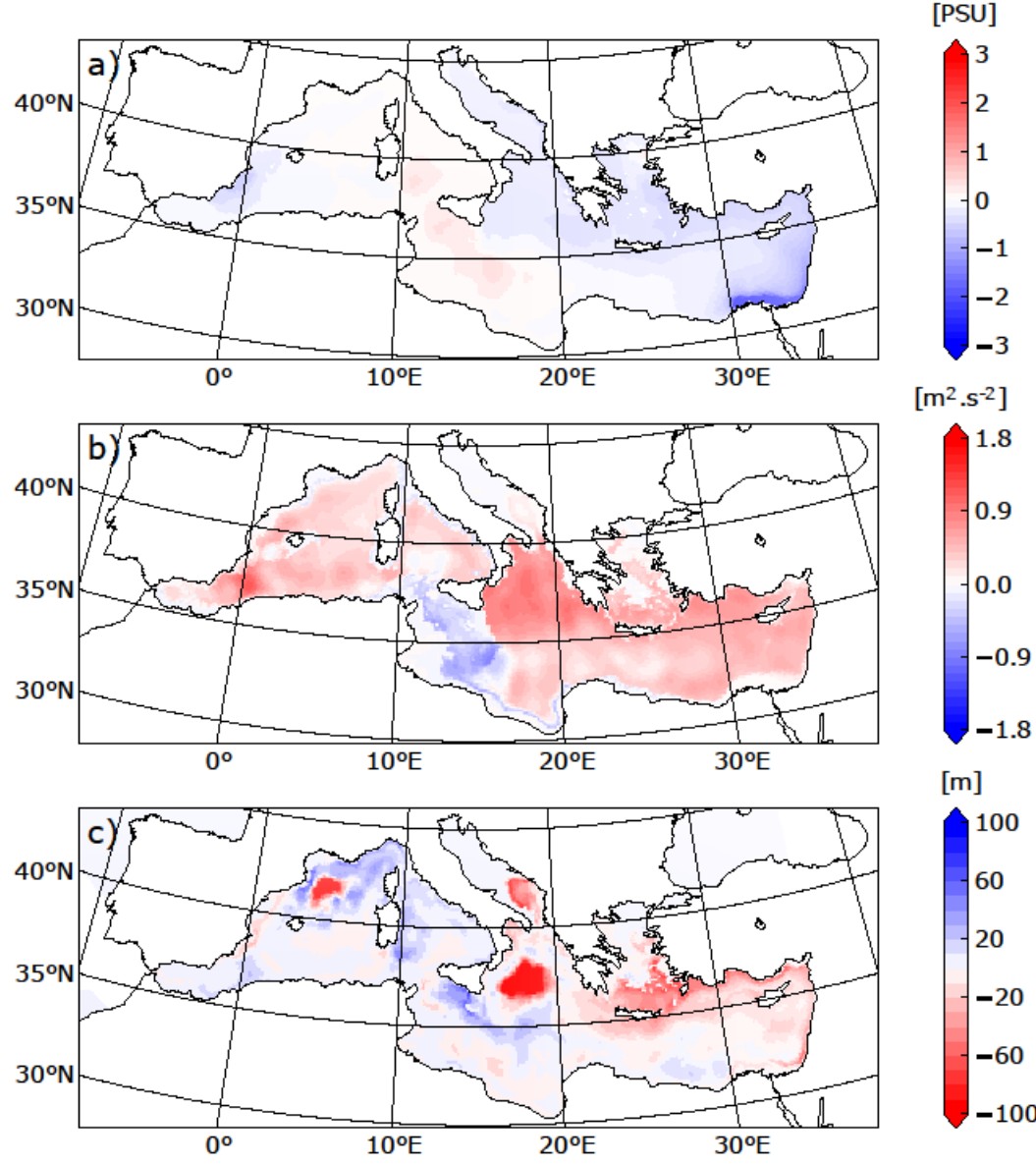


**Figure 12: Deviations between EHOL and PICTRL in a) sea surface salinity, b) index of stratification, c) mixed-layer depth, averaged over the last 30 years of simulation**



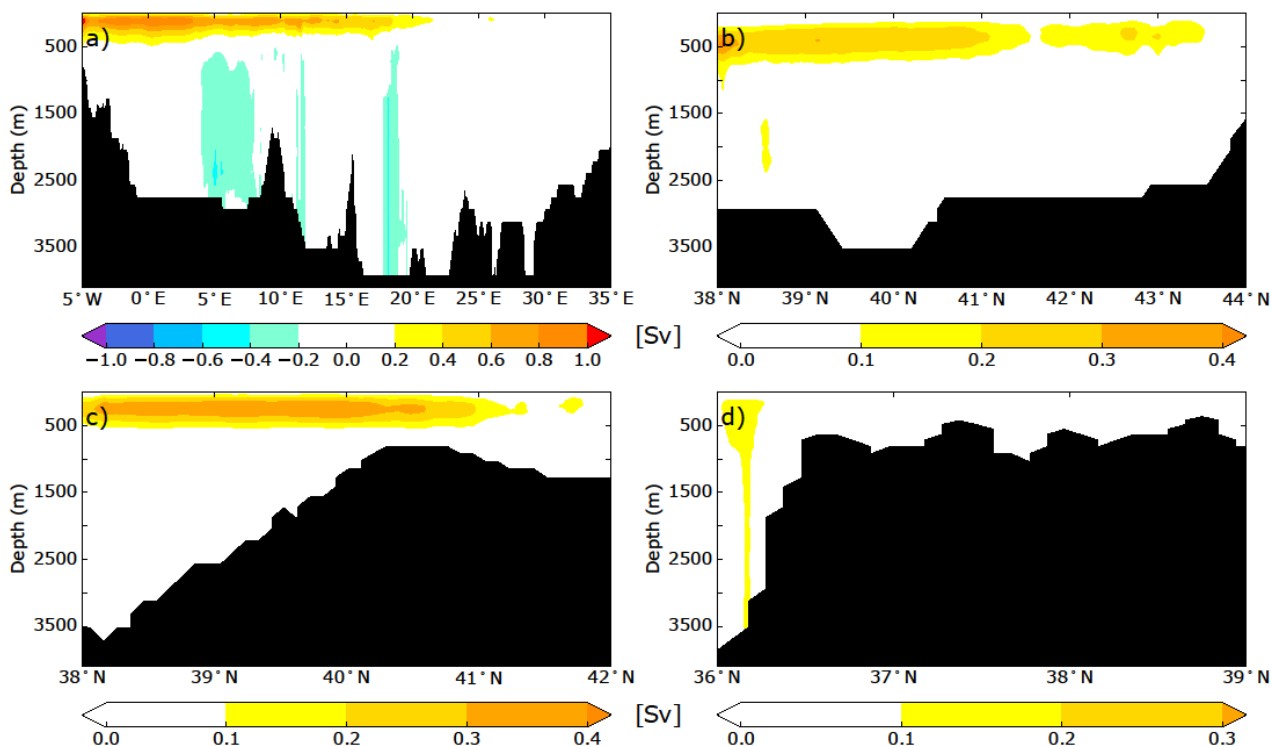

**Figure 13: ZOF (a) and MOF (b, Gulf of Lion, c, Adriatic/Ionian Sea, d, Aegean Sea) for EHOL experiment, averaged over the last 30 years of simulation. These overturning stream-functions were calculated in the same way as in Fig. 6, providing a strict comparison with the experiments HIST and PICTRL.**

## 5    Conclusion and perspectives for the modelling platform

This study aimed to develop a modelling platform to simulate different climatic conditions of the Mediterranean basin. We developed a useful regional climate investigation platform with high spatial resolution over the Mediterranean region. This is particularly relevant for the study of impacts on the circulation of the Mediterranean Sea. The model chain has been evaluated for the historical period. We have presented Early Holocene simulations as an example of the potential of this platform to simulate past climate. For the Early Holocene, our model reproduced satisfactorily the global and regional climate features, compared to the observed data. Our platform allowed, for the first time, the generation of a high-resolution freshwater budget for this period, with a particular focus on continental precipitation, a key factor for the Mediterranean Sea in the assessment of its impact on circulation during the onset of the sapropel event, S1. An important limitation of our sequential approach is the fact that it does not take account of feedback of ocean changes on atmospheric circulation. However, this architecture allows eventual bias correction, conducted at different levels of the platform if needed. One way to overcome

this problem of interactive ocean would be to consider an "asynchronous mode", namely, to take account
of feedback from the ocean component to the atmosphere at a yearly or decadal frequency.


The modelling sequence, moving from global simulation at low resolution to high-resolution regional
ocean modelling, avoids the problem of boundary conditions, and provides a fully consistent platform
that may be used for many paleoclimate studies. We focused here on the Early Holocene period but this
architecture could be used to study other periods investigated in MIPs, such as the Last Glacial
Maximum or the deposition of older sapropels, from the Pliocene to the Quaternary, as long as the
tectonics remain mainly unchanged (PMIP, PlioMIP).


**Code and data availability**. The current version of LMDZ and NEMO are available from the project
website: https://forge.ipsl.jussieu.fr/igcmg_doc/wiki/DocImodelBlmdz and
http://forge.ipsl.jussieu.fr/nemo/wiki/Users under the terms of the CeCill license for LMDZ and
NEMO both. The exact version of the model used to produce the results used in this paper is archived
on Zenodo (Vadsaria et al., 2019), as are input data and scripts to run the model and produce the plots
for all the simulations presented in this paper.

**Author's contribution.** This study was co-designed and approved by all co-authors. The simulation
protocol was constructed by TV and LL from a modelling architecture provided by LL. TV conducted
the numerical simulations and drafted the first version of the manuscript. All co-authors are largely
involved in the writing and revision of the manuscript.

**Acknowledgments.** We thank Mary Minnock for her professional English revision. This work was
supported by the French National program LEFE "HoMoSapIENS". This work was granted access to
the HPC resources of TGCC under the allocation 2017-A0010102212, 2018-A0030102212 and 2018-
A004-01-00239 made by GENCI.

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
