# Peer review of "Development of a sequential tool, LMDZ-NEMO-med-V1, to conduct global to regional past climate simulation for the Mediterranean basin: An Early Holocene case study"

_Geoscientific Model Development, 2019_

## Referee Comment (RC1) · Anonymous Referee #1 · 26 Sep 2019

Review of the ms. 'Development of a sequential tool, LMDZ-NEMO-med-V1, to conduct global to regional past climate simulation for the Mediterranean basin: An Early Holocene case study' by Tristan Vadsaria, Laurent Li, Gilles Ramstein and Jean-Claude Dutay submitted to GMD

The paper describes a technical tool/technique to perform a dynamical down-scaling for the Mediterranean from a simulation with a global climate model. This is applied to a preindustrial/historical simulation and an early Holocene climate state. Whereas some aspects might be useful for other model systems as well, the focus lies on the models

used at LMD: LMD atmosphere (global and regionally zoomed) and the Mediterranean setup of NEMO. The usefulness of the technique is mostly demonstrated for the early Holocene.

The general approach (global AOGCM/ESM -> global AGCM driven with SST and SIC (sometimes with flux/bias corrections) -> regional ARCM -> regional Mediterranean OGCM) is fairly standard for evaluating future (and recent) climate changes for a regional ocean domain. However, this typically involves quite some handwork. The new aspect here is that there is an automatic procedure that simplifies the handling of this model chain. The authors apply this model chain also to the early Holocene, where a downscaling using a regional ARCM to my knowledge has not been attempted before.

In general, the text reads well, there are, however, some problems with the figures, where a more thorough proofreading would have been useful.

The nomenclature should be unified as well. As an example, in the text and the figures/captions sometimes LMD-global/regional, sometimes AGCM/ARCM (e.g. figs. 8/9 and 7) is used.

From the description of the set up, it is not clear, whether the upper boundary conditions for the OGCM does include some restoring-term to a prescribed SST field in addition to the prescribed heat fluxes. This is important, as this seriously affects the interpretation of simulated SST signals. This needs to be clarified in the ms.

The analysis of the Early Holocene simulation is a bit superficial, but this simulation acts rather as a proof of concept, so this is not a major problem. In some plots I had troubles to find the signals the authors were mentioning, some plots might even be wrong.

In general, I believe that quite some revisions are necessary before the paper can reach a state sufficient for publication in GMD.

Detailed comments:

abstract

Please explain to what extent this paper is useful to readers not using the LMD model system.

l32

'it' obviously is supposed to refer to Mediterranean basin, but this seems not to be backed by the structure of the sentence.

?seat? of civilizations.

l55

'In this paper, we developed' -> Here we describe

l94

'surface fluxes and wind stresses from observations' I am not aware of daily observational data for fluxes.. You probably refer to reanalysis products. With respect to fluxes, these include the use of a model and its parametrisations. Please be correct. same in l115.

l101

'the method is not well adapted'

In fact it is and superior to what you propose, the only problem is the computational effort for long simulations.

l120

it is possible but rather expensive. Please be more specific.

l152

An alternative could be to rerun the coupled model with high frequency output for 30 years rather than to rerun an AGCM. please discuss.

l191

1.875°x1.25° with 96x72 grid points (from Fig.1) does not result in a global domain! Please correct.

l197

Please specify the frequency of the required AGCM output

l201

please specify in this section, whether any restoring of SST to prescribed values is involved. Is there any flux correction for P-E?

l275 /Fig.2 caption

define Mediterranean region/ Mediterranean-only. Does this mean only over the ocean?

l278

to what extent is the the response in 2m-air temperature over the ocean surprising if the SST is prescribed? Would you have gotten the same trend from the global AOGCM?

Fig. 3b

It seems that the anomalies HIST-OBS in panel b are not anomalies but the same fields as in 3a except with a different colour bar and masking over the ocean. Here you use HIST-OBs as anomalies, in Fig. 2 obviously as absolute values. Please stick to one definition.

l293

BASIN MEAN 'P and E over the Med Sea ARE very close ...'

please correct!

Table 3

please include an extra column with the total freshwater budget of the Med (saves the reader from doing it him/herself).

Figure 5

MLD averaged over the entire year is not very useful. Rather use annual max MLD or winter (Feb or March) MLD. This would indicate the depth of convection and thus the locations of deep water formation. This would fit to your use of this figure in l336. give Table 1

bias of a simulation would be HIST-obs. From Fig. 4 I conclude that the model is too cold and salty. Here you seem to use a different sign for bias, which is confusing for the reader.

l336

thicker -> deeper Please explain, why the simulated MLD is deeper in the EMed.

Fig. 6

why do we see in the ZOF deep cells both in EMed and WMed > 0.2 Sv but no corresponding water mass movement in the Gulf of Lions and the Adriatic? The deep branches seem to be < 0.1 Sv. Please explain this. Specify the longitudinal extent of the domains used to calculated the MOFs. The topography in the Adriatic MOF seems to be pretty deep, please check. You are using rows/columns in a wrong way. Where in Fig. 6 is the 3rd column from left, there are only 2 columns. (should be row from top) Please correct.

l348+

There must be more simulations than just the ones using the same ocean model setup. There are more models, e.g. the MIT model. Are there any estimates from observations?

Please compare:

l350: A large spread between the models for this pattern indicates that there is still a lack of modelling capacity to simulate the deep circulation of the Mediterranean Sea.

l367: The thermohaline circulation is well captured by the oceanic model (compared to the simulations of Adloff et al., 2015 and Somot et al., 2006 for instance), which inspires confidence in our modelling platform for the investigations of past climate.

For me these two statements do not go together very well...

Figure 7

Please include labels a), b) etc. The top right panel looks like summer temperatures, but has a colour bar indicating mm/d. Inverse problem in bottom left panel. Please use same colour bar for summer and winter temps.

Compare Figs. 7 and 10!

Assuming that LMD-Global is equal AGCM, why is Europe so much drier in Fig. 10 than in Fig. 7? Shouldn't these panels show the same signals? Please explain.

Fig. 10

Please use the same colour bar in all panels!

Fig. 9

Please add arrows in AGCM plots.

Fig. 11

A mess! Split it up into 2 figs. and make sure that there is a clear relation between colour labels and displayed data panel. Why is the Nile shown in the west as well? If the Nile is flux corrected in EHOL, how can there be an anomaly of <-3000 during winter. Does this indicate a negative Nile runoff in EHOL winter? Please explain and discuss implications (deep convection in Nile plume?).

l530 and Fig. 12c

Please change consistent with Fig. 5!

l521

Please indicate in this section, how close the surface is to steady state. Please show time series of basin mean SSS during the EHOL and PICTRL simulations. Maybe in the supplement.

Fig. 13

Please correct the caption Ionian should be Aegean.

l530

Comparing Figs 6 and 13 it seems that the ZOF in EHOL is about as strong as in HIST. Compared to PICTRL it is indeed reduced. In the MOFs it is hard to see the reduction which is claimed to be obvious ('is followed by a general reduction in the thermohaline circulation compared to PICTRL'). Please make a careful and more detailed comparison. And include discussion of Fig. S7 which shows only a weak reduction.

l579

you also used preindustrial pCO2 instead of early Holocene pCO2, which should about 260 ppm. Please mention.

supplement

l180

'latest version'

not a particular good description, especially in a few years from now. Specify the version.

l260

Please mention that the method can lead to negative river runoff. Is this then effectively

the same as a very strong local evaporation? Does this initiate salt driven convection at the mouth of the Nile?

l299

Please compare the results shown in Fig.S2 with the bias corrected SST used to drive the global AGCM. Is there a real improvement or do you get more or less the same results? Compare with similar plots in Mikolajewicz (2011), who got almost no difference in the simulated climate signal.

―――――――――――――

---

## Referee Comment (RC2) · Anonymous Referee #2 · 4 Nov 2019

Review of GMD-2019-196 Vadseria et al. present a sequential modelling tool to investigate (paleo-)climate change effects on Mediterranean Sea circulation. They first explain their set-up and validate for the present-day. Then an example of application, the Early Holocene, is given. It seems like a valid approach that is indeed of use for multiple (paleo-) applications. I would however suggest revision to make the paper clearer, both structurally and with respect to what exactly the added value of their sequential modelling tool is. So my main comments are:

- structurally the paper can improve to clear up some unclarities. For instance, Fig.

2 states "hist-obs" while the text only mentions "hist". I guess you mean the same simulation. Also, many citations seem to be absent from the reference list.

- content-wise, the authors seem to claim that high-resolution atmospheric forcing is needed to get correct Mediterranean Sea circulation. This needs to be better substantiated by results or discussion. For instance, can you show that your simulation yields better results than, say, a OGCM run forced directly with AGCM forcing rather than ARGCM?

Please find more detailed comments below, followed by the GMD review criteria.

P2, line 67 "the localization of the ... of debate": true, and actually your set-up would allow for testing separate forcing sources for sapropel formation (i.e. only adding additional freshwater to a certain location, or only precipitation versus only river runoff). This would make your model setup even more useful than using it for overall Med-Sea circulation under paleo-climate-forcings.

P3 lines 73-77. Please provide section numbers when outlining the paper.

P4 lines 130-140: how about the exchange with the Black Sea? Is it common to deal with as if a river?

P5 section 1.3: in my opinion this fits better in the methods section, where it can be merged with the specific LMDZ-NEMO set-up.

P6 lines 188-190: mention where it can derive boundary conditions from (SIC and SST).

P6 lines 199-200: give a reference for ORCHIDEE and is it run at the same resolution?

P6 line 208: which 'first dataset of river discharges' do you refer to? And does this represent the majority of discharge in the 192 ORCHIDEE river mouths?

P7 lines 211-213: how realistic is the assumption that water from the Black Sea is fresh? And does the Q+P-E budget over the Black Sea derive from the AGCM or

ARCM?

P7 line 215 / fig 1: to fit the figure with all your simulations, can you include that boundary conditions can also derive from reanalysis?

P7 line 229: maybe put the table that shows an overview of experiments in the main text.

P7 line 239: the cited paper is not in the reference list (as are many other citations)

P8 line 246: "for one period" rather than "for a period"

P8 Fig 1: usually u is zonal wind, v is meridional wind.

P8 line 266: write out WOA

P9 Fig 2: the legend mentions "HIST-OBS", I guess you mean experiment "HIST"? Also, why do you use ERA20C here whereas experiment "HIST" is forced with ERA-Interim?

P10 line 291: Table 2, not 3

P10 Fig 3: again a different dataset is used (CRU), whereas Fig. 2 compares to ERA20C, and "HIST" is forced with ERA-Interim. Why would you use such a range of datasets? And why not use a reanalysis that has values over the sea? Also, looking at the color scales, it seems that the overestimation is as large as the modelled precipitation itself over land. So the relative overestimation there is near 100%?

P11 Fig 5: in the upper panel it seems like there is a contour overlaying the colours, are those from observations?

P12 Table 1: provide units and define IS.

P12 line 337: refers to 5b, instead of 5a?

P12 line 340: Figure 6a instead of 7a.

P13 lines 350-352: if there is still a lack of modelling capacity to simulate Med-Sea deep circulation, how can you verify that your study is an improvement?

P14 lines 362-364: Figures 2 and 4 show that your simulations results in significantly lower temperatures than observed, yet here you say they are consistent?

P14 line 365: How can a model overestimate the precipitation over the surrounding land substantially (fig 3) yet have precipitation over the sea close to observation (Table 2) and have lower river runoff than HIST or PICTL (with overestimation of precipitation over land, why is runoff not overestimated too – is this due to bias correction?)

P15 section 3.2: is there any additional ice sheet remaining in the early Holocene in the model?

P16 line 398-399: "increased Early Holocene summer insolation" or "increased Early Holocene insolation seasonality".

P16 line 400-404: refer to figures 7c, 7d.

P17 Figure 7: in the caption the "a" after "b) summer temperatures" should be removed

P20 line 494-497: how does the increased nile runoff in PICTRL (do you mean compared to observations?) compare to the overall lower runoff reported in table 2?

P21 Fig 11: Especially in late winter and summer, runoff from the Black Sea is decreased by roughly the same order of magnitude as the increase in Nile runoff. Can you reflect on the possible role that the Black Sea runoff alone could have in sapropel formation?

P22, lines 522-525: what do you mean by the reference for correction is the pre-industrial state? How is river runoff corrected based on pre-industrial climate?

P22 lines 543-545: I would not say that your simulations shows similar changes as Adloff or Bosmans. For instance Adloff (their fig 9) shows strong salinity increases around Greece, and Bosmans (their fig 11) do not show a decreased mixed layer depth

in the Ionian sea.

P24 Fig 13: add to caption that this can be compared to Fig 6 (PICTRL).

P24 line 562: "for the first time" – this you could mention more clearly in the introduction.

P25 lines 571-579: this is not a section that should be in the Conclusions. It is more fitting for a discussion section. It also makes me wonder if there is anything known of the effect of keeping the Bosphorus exchange as it is today.

P26 References: make sure all cited literature is in the reference list.

———————————————————————————————— 1. Does the paper address relevant scientific modelling questions within the scope of GMD? Does the paper present a model, advances in modelling science, or a modelling protocol that is suitable for addressing relevant scientific questions within the scope of EGU?

Yes. Protocol is suitable for addressing paleo-questions. The models themselves are however not new, and I would like to see more evidence that the modelling chain used in this paper is necessary (e.g. compare to a NEMOMED8 simulation directly forced with AGCM instead of ARCM) and / or can be used to address questions not yet addressed (e.g. separate the freshwater flux changes. Ocean simulations of e.g. Holocene or LGCM are not new.)

2. Does the paper present novel concepts, ideas, tools, or data? The specific modelling chain (AGCM -> ARCM -> OGCM) seems new for the Mediterranean.

3. Does the paper represent a sufficiently substantial advance in modelling science? I think so. I would, however, like to see this more substantiated (see Q1)

4. Are the methods and assumptions valid and clearly outlined? Valid yes. For clarity it would be good to put a table overview of all simulations in the main text (I found it in the supplementary though)

5. Are the results sufficient to support the interpretations and conclusions? The model

set-up is sufficiently clear, but whether this particular set-up is needed (rather than e.g. just using global GCM directly onto a Med-OGCM) could be more substantiated.

6. Is the description sufficiently complete and precise to allow their reproduction by fellow scientists (traceability of results)? In the case of model description papers, it should in theory be possible for an independent scientist to construct a model that, while not necessarily numerically identical, will produce scientifically equivalent results. Model development papers should be similarly reproducible. For MIP and benchmarking papers, it should be possible for the protocol to be precisely reproduced for an independent model. Descriptions of numerical advances should be precisely reproducible. Yes.

7. Do the authors give proper credit to related work and clearly indicate their own new/original contribution? Credit is given (although a lot of references in the text are missing from the reference list!!), and the specific modelling chain seems new for the (paleo-) Mediterranean.

8. Does the title clearly reflect the contents of the paper? The model name and number should be included in papers that deal with only one model. Yes.

9. Does the abstract provide a concise and complete summary? Yes.

10. Is the overall presentation well structured and clear? No, I think the paper could benefit from some clarifications. It is sometimes unclear which simulations are referenced to and what simulations' boundary conditions are (For instance in Figure 2 HIST-OBS is not yet defined). An overview of experiments in the main text (as opposed to only in the supplementary material) would be helpful. Also I feel part 1.3 is too overlapping with the methods, as it essentially a generic version of the methods. The distinction between results-discussion-conclusion also needs to be revised. For instance, the conclusion contains a paragraph on how the Bosporus was dealt with, which should be in discussion. Furthermore I found that many citations in the text are not present in the reference list.

11. Is the language fluent and precise? yes.

12. Are mathematical formulae, symbols, abbreviations, and units correctly defined and used? N.a.

13. Should any parts of the paper (text, formulae, figures, tables) be clarified, reduced, combined, or eliminated? I think section 1.3 could be removed or combined with the methods section (see Q10). There should also be a clearer overview of the simulations.

14. Are the number and quality of references appropriate? Yes, but do make sure all citations are actually present in the reference list.

15. Is the amount and quality of supplementary material appropriate? For model description papers, authors are strongly encouraged to submit supplementary material containing the model code and a user manual. For development, technical, and bench-marking papers, the submission of code to perform calculations described in the text is strongly encouraged. I appreciate the links to the model code as well as the user manual in the supplementary materials.

---

## Editor Decision (ED1)

**Requested minor revisions to gmd-2019-196**

L228: Clarify the procedure for boundary conditions applied when running NEMO ocean only model. This paragraph is currently too vague and It's difficult to make sense of it. Clarify what the flux are, what the restoring term is and what the constant coefficient controls. If there is a paper you can cite that would be great, otherwise please include an equation.

L306: "The global simulation, after SST bias correction, ranged with the observation, compared to IPSLCM5A (Figure 2)." "ranged with the observation" is not clear. Do you mean "has the same range of variability as e.g. the 2m temperature over the Mediterranean region from ERA20C " ?

L219 "A first dataset of climatological river discharges is proposed by default to cover the entire Mediterranean draining basin with represents 33 river mouths." What do you mean by "is proposed" to whom, for what? How is it used? Please clarify the text.

"[Reviewer comment]
P7 lines 211-213: how realistic is the assumption that water from the Black Sea is fresh?
And does the Q+P-E budget over the Black Sea derive from the AGCM orARCM?
[Reply]
It is a commonly-used treatment when the Mediterranean model doesn't include the Black Sea. The fresh water assumption is entirely justified although the actual water flow from the Black Sea can be salty, since what we evaluated in terms of E, P and Runoff is indeed the fresh water budget. What is important in the model is not the water mass itself, but the salt content. We made some revisions in the new manuscript for this regard."
I can't see how and where you have addressed this point. Please clarify with citation of the text.

"[Reviewer comment]
P14 lines 362-364: Figures 2 and 4 show that your simulation results in significantly lower temperatures than observed, yet here you say they are consistent?
[Reply]
Yes, there are cold biases. We changed the corresponding text in the revised manuscript "The atmospheric simulation is acceptable compared with observations for the air temperature at 2m at both global and regional scales "(l405)."
I am not satisfied with how you have addressed this point. Changing the word "consistent" with "acceptable" just makes a wrong statement into something vague and subjective. Please quantify and describe the cold bias and discuss the implications here.

"[Reviewer comment]
P22, lines 522-525: what do you mean by the reference for correction is the preindustrial state? How is river runoff corrected based on pre-industrial climate?
[Reply]
We choose to "correct" the Mediterranean river runoff during the Early Holocene based on the precipitation difference (EHOL – PICTRL) coming from both the ARCM and AGCM and apply it to the PICTRL river runoff (which was prescribed). The procedure of river runoff is detailed in the supplementary material (**Text S2: Bias correction**)"

This point hasn't been addressed adequately, please correct the text when you make reference to the "correction" to clarify that you apply a bias correct as described in Text S2.

Editorial corrections (suggested changes in bold):
L 153: "This architecture is based on a method **that provides** as much **compatibility** as possible among**st** the models used and high **consistency** with data."

L224L "river mouths **that** cover the …"

L330 " both the precipitation and evaporation over the Mediterranean Sea in HIST **are** very close to the observations" Quantify how close.

L331 "**The** two other simulations **included in Table 1**, PICTRL and EHOL, are those designed to investigate**s** the Early Holocene climate **(see Section 4)**."

L407: "The ZOF **in HIST depicted in Figure 6)**"

Figure 6: replace row numbers with figure labels (a-h)

L425 It is not correct to say "ranges with the observation". Do you mean "has the same range of variability as the observations"

---

## Author Response (AR3)

**Reply to Anonymous Referee #1**

[Reviewer comment]
The paper describes a technical tool/technique to perform a dynamical down-scaling for the Mediterranean from a simulation with a global climate model. This is applied to a preindustrial/historical simulation and an early Holocene climate state. Whereas some aspects might be useful for other model systems as well, the focus lies on the models used at LMD: LMD atmosphere (global and regionally zoomed) and the Mediterranean setup of NEMO. The usefulness of the technique is mostly demonstrated for the early Holocene. The general approach (global AOGCM/ESM -> global AGCM driven with SST and SIC (sometimes with flux/bias corrections) -> regional ARCM -> regional Mediterranean OGCM) is fairly standard for evaluating future (and recent) climate changes for a regional ocean domain. However, this typically involves quite some handwork. The new aspect here is that there is an automatic procedure that simplifies the handling of this model chain. The authors apply this model chain also to the early Holocene, where a downscaling using a regional ARCM to my knowledge has not been attempted before. In general, the text reads well, there are, however, some problems with the figures, where a more thorough proofreading would have been useful.

[Reply]
We thank the reviewer for the careful and detailed review, and also for the numerous constructive advices. All of them were carefully implemented in the revised manuscript.

[Reviewer comment]
The nomenclature should be unified as well. As an example, in the text and the figures/captions sometimes LMD-global/regional, sometimes AGCM/ARCM (e.g. figs. 8/9 and 7) is used.

[Reply]
We agree that the initial manuscript was confusing for this aspect. In fact, we use AGCM / ARCM when we describe the general aspect of our approach, and we use LMDZ4-global / LMDZ4-regional when evoking the actual implementation of the modelling chain. We improved this aspect throughout the manuscript, including main text and figure captions.

[Reviewer comment]
From the description of the set up, it is not clear, whether the upper boundary conditions for the OGCM does include some restoring-term to a prescribed SST field in addition to the prescribed heat fluxes. This is important, as this seriously affects the interpretation of simulated SST signals. This needs to be clarified in the ms.

[Reply]
When the oceanic model NEMO is used alone, with prescribed surface fluxes, it is indispensable to implement a restoring term with a constant coefficient of 40 W/m2/K. This is a standard procedure for NEMO to prevent eventual run-away cases. In our modelling chain, the target temperature for the restoration is the surface air temperature from the regional atmospheric model.

[Reviewer comment]
The analysis of the Early Holocene simulation is a bit superficial, but this simulation acts rather as a proof of concept, so this is not a major problem.

[Reply]
Thanks, indeed we chose to publish our platform in GMD and to provide just an illustration. This is why we do not emphasize too much on this case study. Nevertheless, as also suggested by the second reviewer, we provide more information on the improvements obtained going from global to regional scales.

[Reviewer comment]
In some plots I had troubles to find the signals the authors were mentioning, some plots might even be wrong. In general, I believe that quite some revisions are necessary before the paper can reach a state sufficient for publication in GMD.

[Reply]
We believe that the revised manuscript is improved for this aspect. Many thanks to the reviewer for his/her careful reading of the manuscript We improved the plots, clarified most of them and corrected the errors raised by the reviewer.

[Reviewer comment]
Detailed comments:
Abstract        Please explain to what extent this paper is useful to readers not using the LMD model system.

[Reply]

(All the lines mentioned hereafter refer to new version of the manuscript)

This manuscript is a compromise between a general concept of a sequential modelling platform from global to regional and an actual implementation with numerical tools available in IPSL. We hope that our manuscript can help to promote such an approach in dealing regional climate issues. The concept that we proposed can be easily extended to other groups with a similar background and a focus on high-resolution climate modelling

 [Reviewer comment]
Line 32        'it' obviously is supposed to refer to Mediterranean basin, but this seems not to be backed by the structure of the sentence. ?seat? of civilizations.

[Reply]
We rephrased this sentence. By "seat of civilizations", we meant that the Mediterranean basin played an important historic role for human civilizations.

[Reviewer comment]
l55      'In this paper, we developed' -> Here we describe

[Reply]

We rephrased this sentence and a related sentence later in the paragraph.

[Reviewer comment]
l94    'surface fluxes and wind stresses from observations'
I am not aware of daily observational data for fluxes. You probably refer to reanalysis products. With respect to fluxes, these include the use of a model and its parametrisations. Please be correct. same in l115.

[Reply]
Yes, we agree, it is almost impossible to have good observation of fluxes at air-sea interface for daily frequency and for large extension in space. Such fluxes can only be obtained within a model (for climate simulation or meteorological analysis and re-analysis). We made the necessary modification accordingly (l100).

[Reviewer comment]
l101
'the method is not well adapted'
In fact it is, and superior to what you propose, the only problem is the computational effort for long simulations.
l120
it is possible but rather expensive. Please be more specific.

[Reply]
Yes, we agree. We recognize that our general framework of a sequential modelling chain is also a way to remediate the issue of computational resources. We made the necessary modification accordingly (l130).

[Reviewer comment]
l152    An alternative could be to rerun the coupled model with high frequency output for 30 years rather than to rerun an AGCM. please discuss.

[Reply]
Yes, we agree. The best way is to save the high-frequency 3-D outputs when running (or re-running) the paleo applications in their fully-coupled configuration. But this is not always possible or easily feasible. Our proposition of running only AGCM (the same one as in the coupled-mode application, or another independent one) is based on pragmatic consideration. Previous studies (references cited in the main manuscript), including some of our own studies, seem to validate this approach using AGCM with climate signals from SST and SIC in ocean-atmosphere coupled models.

[Reviewer comment]
l191    1.875∘x1.25∘ with 96x72 grid points (from Fig.1) does not result in a global domain! Please correct.

[Reply]
We apologize for the mistake. That should be 3.75 in longitude and 2.5 in latitude (l197).

[Reviewer comment]
l197    Please specify the frequency of the required AGCM output l201

[Reply]
We now provided these parameters: AGCM to ARCM every 2 hours; Fluxes from ARCM to ORCM every day; Atlantic buffer zone and discharges from rivers updated every month (l203).

[Reviewer comment]
L201   please specify in this section, whether any restoring of SST to prescribed values is involved. Is there any flux correction for P-E?

[Reply]
There is no flux correction in running our oceanic model NEMO: neither heat fluxes, nor water fluxes (P and E), nor wind stress. However, when the oceanic model NEMO is used alone, with prescribed surface fluxes, it is indispensable to implement a restoring term with a constant coefficient of 40 W/m2/K. This is a standard procedure for NEMO to prevent eventual run-away cases. In our modelling chain, the target temperature for the restoration is the surface air temperature from the regional atmospheric model. We made the necessary modification accordingly (l222).

[Reviewer comment]
l275 /Fig.2    caption define Mediterranean region/ Mediterranean-only. Does this mean only over the ocean?
l278    to what extent is the the response in 2m-air temperature over the ocean surprising if the SST is prescribed? Would you have gotten the same trend from the global AOGCM?

[Reply]
We apologize for the confusion. In fact, for both global average or Mediterranean average, we used surface air temperature at 2 m from land and water bodies. The Mediterranean average corresponds to the regional domain of LMDZ4-regional. We improved the caption to avoid any confusion. During the revision, we also added the global T2m from the ocean-atmosphere coupled model IPSL-CM5A, considered as a baseline, in order to appreciate the improvement that we made in our system (Figure 2 and l298).

[Figure]

New figure 2: **Time series of annual mean surface air temperatures at 2 m in HIST (red) and ERA20C (black) and IPSLCM5A (green) for global average (solid lines) and Mediterranean-region (ocean and continent) average (dashed lines).**

[Reviewer comment]
Fig. 3b It seems that the anomalies HIST-OBS in panel b are not anomalies but the same fields as in 3a except with a different colour bar and masking over the ocean. Here you use HIST-OBs as anomalies, in Fig. 2 obviously as absolute values. Please stick to one definition. l293 BASIN MEAN 'P and E over the Med Sea ARE very close ...' please correct!

[Reply]
Thanks for your careful reading. Indeed, the initial panels in Fig. 3 were not very relevant for our purpose to illustrate the performance of our platform in simulating the rainfall. We finally changed Fig. 3 to a new illustration in the form of zonal and meridional averages (including more results). We also changed the description in the manuscript, accordingly (see section 3.3).

[Figure]

**New Figure 3: Annual mean precipitation, a) meridionally averaged (30 to 50°N), b) zonally averaged (-10 to 35°E), in the historical simulations with AGCM (LMDZ-global) and ARCM (LMDZ-regional). Observation comes from GPCP (Global Precipitation Climatology Project, 1979 to 1999, blue line, ref: Adler et al., 2018). and ERA20C (green line, ref: Stickler et al., 2014).**

[Reviewer comment]
Table 3        please include an extra column with the total freshwater budget of the Med (saves the reader from doing it him/herself).

[Reply]
Thanks. This is Table 1 now in the revised manuscript. We now completed it, including all terms of the fresh water budget over the Mediterranean Sea.

[Reviewer comment]
Figure 5        MLD averaged over the entire year is not very useful. Rather use annual max MLD or winter (Feb or March) MLD.This would indicate the depth of convection and thus the locations of deep water formation. This would fit to your use of this figure in l336. Table 1 bias of a simulation would be HIST-obs. From Fig. 4, I conclude that the model is too cold and salty. Here you seem to use a different sign for bias, which is confusing for the reader.

[Reply]
We apologize for the confusion. The figure caption was not appropriate. Our diagnostics were indeed the winter maximum value of the mixed layer depth. We corrected it accordingly in the revised manuscript. In Table 2 (initially labelled Table 1), we now corrected the sign of the convention.

[Reviewer comment]
l336    thicker -> deeper Please explain, why the simulated MLD is deeper in the EMed.

[Reply]

We think that a thicker MLD in the eastern basin is due to the salty conditions.

[Reviewer comment]
Fig. 6 why do we see in the ZOF deep cells both in EMed and WMed > 0.2 Sv but no corresponding water mass movement in the Gulf of Lions and the Adriatic? The deep branches seem to be < 0.1 Sv. Please explain this. Specify the longitudinal extent of the domains used to calculated the MOFs. The topography in the Adriatic MOF seems to be pretty deep, please check. You are using rows/columns in a wrong way. Where in Fig. 6 is the 3rd column from left, there are only 2 columns. (should be row from top) Please correct.

[Reply]
We firstly corrected the issue of row/column confusion and we also detailed the domain used for our calculation of the overturning stream function. The Adriatic MOF seems deeper, since our calculation includes the north of the Ionian Sea. But our MOF roughly corresponds to those of other similar studies (e.g. Somot et al. 2006 Fig. 11 and Adloff et al. 2016 Fig. 6). ZOF being integrated from the south coast to the north coast, and MOF from the west coast to the east coast for a particular semi-closed sector, we can observe different deep cells in the ZOF and MOFs. In fact, ZOF includes the circulation near the African coast which is in none of the MOFs

[Reviewer comment]
l348+ There must be more simulations than just the ones using the same ocean model setup. There are more models, e.g. the MIT model. Are there any estimates from observations? Please compare:

[Reply]
Under the Med-CORDEX framework, there are some initiatives for inter-comparison of models over the Mediterranean area. Results and publications are expected soon. In the recent literature, we also found an interesting work of Pinardi et al. (2019) who present ZOF derived from their reanalysis data (1987-2013). It seems that our ZOF in HIST is weaker than that from observation. We updated the text accordingly (l393): "The ZOF depicts in HIST simulation is consistent with the reanalysis (1987-2013) of (Pinardi et al., 2019) over the Western basin, but show a weaker Eastern deep cell compared to the reconstruction."

[Reviewer comment]
l350: "A large spread between the models for this pattern indicates that there is still a lack of modelling capacity to simulate the deep circulation of the Mediterranean Sea."
l367: "The thermohaline circulation is well captured by the oceanic model (compared to the simulations of Adloff et al., 2015 and Somot et al., 2006 for instance), which inspires confidence in our modelling platform for the investigations of past climate."
For me these two statements do not go together very well.

[Reply]
However, there is some uncertainties concerning the changes in deep circulation for the Mediterranean Sea. Our simulation is nevertheless in the range of circulation changes provided by different modeling studies. Therefore, the sensitivity for historical period is encouraging to go a step further and to investigate a larger perturbation as the early Holocene one. To remove any confusion, we just deleted the first phrase and made some revisions for the second one (l410).

[Reviewer comment]
Figure 7        Please include labels a), b) etc. The top right panel looks like summer temperatures, but has a colour bar indicating mm/d. Inverse problem in bottom left panel. Please use same colour bar for summer and winter temps.
Compare Figs. 7 and 10! Assuming that LMD-Global is equal AGCM, why is Europe so much drier in Fig. 10 than in Fig. 7? Shouldn't these panels show the same signals? Please explain. Fig. 10 Please use the same colour bar in all panels!

[Reply]
We apologize for the wrong label in Figure 7b. We corrected it now. For the apparent difference between Fig. 7c and Figure 10c, it was mainly due to a small calendar shift, combined with a graphic problem in relation to "contour fill" with python matplotlib. The graphic was now plotted with "shading" option, which seems resolve the problem. We updated both Figs. 7 and 10.

[Figure]

**New Figure 7: Deviations between EHOL and PICTRL in the AGCM for a) winter temperatures at 2m, b) summer temperatures a, c) June to August precipitation, and d) July to September surface runoff (averaged over the entire simulation).**

[Reviewer comment]
Fig. 9   Please add arrows in AGCM plots.

[Reply]
Ok, arrows added now in the revised plots.

[Reviewer comment]
Fig. 11 A mess! Split it up into 2 figs. and make sure that there is a clear relation between colour labels and displayed data panel. Why is the Nile shown in the west as well? If the Nile is flux corrected in EHOL, how can there be an anomaly of <-3000 during winter. Does this indicate a negative Nile runoff in EHOL winter? Please explain and discuss implications (deep convection in Nile plume?).

[Reply]
We apologize for any confusion in Figure 11, and we recognize that it was not an easy graphic to read and to understand. We entirely revised it and made text revisions in the "hydrological changes" subsection. We also split the graphic into two parts, as suggested. We keep only one part in the main text, and we put the second part into Supplementary materials. When we flux-corrected the river runoff there is no negative values, please see the sub section "River runoff to the Mediterranean Sea" of the section "Text S2 bias correction" in the supplementary. See also our response relative to your comment for the supplementary material.

[Reviewer comment]
l530 and Fig. 12c     Please change consistent with Fig. 5!

[Reply]
Yes, we checked the consistency between Figure 12c and Figure 5. They are consistent. We added a phrase in this sense in the revised manuscript (l629).

[Reviewer comment]
l521    Please indicate in this section, how close the surface is to steady state. Please show time series of basin mean SSS during the EHOL and PICTRL simulations. Maybe in the supplement.

[Reply]
We believe that our simulation PICTRL and EHOL reached their stationary state, at least for surface properties. Figures S6 to S8 display the time series for the index of stratification, the zonal overturning stream-function and sea-surface salinity, which allow us to conclude the quasi-stationarity of the simulations. The following panel reproduces Figure S9 showing the evolution of SSS in PICTRL and EHOL.

[Figure]

**Figure S9**: Interannual evolution of the sea surface salinity (SSS) for the Mediterranean Sea for the PICTRL and EHOL simulations (including the PTCRL spin-up phase).

[Reviewer comment]
Fig. 13         Please correct the caption Ionian should be Aegean.

[Reply]
Corrected

[Reviewer comment]
l530    Comparing Figs 6 and 13 it seems that the ZOF in EHOL is about as strong as in HIST. Compared to PICTRL it is indeed reduced. In the MOFs it is hard to see the reduction which is claimed to be obvious ('is followed by a general reduction in the thermohaline circulation compared to PICTRL'). Please make a careful and more detailed comparison. And include discussion of Fig. S7 which shows only a weak reduction.

[Reply]
We were limited to a visual inspection in the manuscript, since this GMD manuscript was mainly devoted to the introduction and presentation of the modelling platform. Detailed diagnostics will come in future works. Nevertheless, if we plot the difference EHOL-PICTRL and EHOL-HIST (as shown here in this review-reply text), we do clearly see the reduction of the Mediterranean overturning circulation.

[Figure]

**Additional figure 1: Overturning stream function. First column: EHOL, second column: EHOL minus PICTRL. From top to bottom are ZOF for the entire**

Mediterranean, MOF for the Gulf of Lion, MOF for the Adriatic/Ionian Sea, and MOF for the Aegean Sea.

[Figure]

**Additional figure 2: The same as in the precedent, but for EHOL-minus-HIST.**

[Reviewer comment]
l579    you also used preindustrial pCO2 instead of early Holocene pCO2, which should be about 260 ppm. Please mention.

[Reply]
This comment is perfectly true. We should have changed in our case study the pCO2 value to 260 ppmv, as it is recommended by PMIP for mid-Holocene. Howerver our **goal in this paper was mainly to have a sensitivity to orbital parameters.** This is clearly stated in the supplementary information section Table S2.

[Reviewer comment]
supplement l180        'latest version' not a particular good description, especially in a few years from now. Specify the version.

[Reply]
Yes, that's right. We deleted the irrelevant words in the revised manuscript (l37 and l152).

[Reviewer comment]
supplement l260        Please mention that the method can lead to negative river runoff. Is this then effectively the same as a very strong local evaporation? Does this initiate salt driven convection at the mouth of the Nile?

[Reply]

Theoretically, a negative river runoff can happen with the water budget treatment in our modelling chain. It would be equivalent to a strong evaporation that can eventually induce a salt-driven convection. But in our case, EHOL shows a general increase of fresh water discharge in comparison to PICTRL, which prevents negative runoff from occurring.

[Reviewer comment]

supplement l299      Please compare the results shown in Fig. S2 with the bias corrected SST used to drive the global AGCM. Is there a real improvement or do you get more or less the same results? Compare with similar plots in Mikolajewicz (2011), who got almost no difference in the simulated climate signal.

[Reply]

We understand your concern and our results confirm your guess. What shown in Fig. S2 (now S3 in the revised manuscript) is the SST in the simulation EHOL, with comparison to a few reconstruction data. You asked if it is consistent with the bias-corrected SST (original SST from IPSL-CM5A, but with biases corrected) that was used to drive both AGCM and ARCM. The answer is Yes. The two fields are quite close to each other, although they do have different spatial resolutions and they differ in detailed structures. Uve Mikolajewicz, in 2011, published a similar study on the Mediterranean Sea climate during LGM (Climate of the Past, doi: 10.5194/cp-7-161-2011). He pointed out (Fig. 15, there) that the SST changes obtained in the regional ocean simulation is very close to those from the initial Earth System Model (MPI-ESM) serving as a driver with an AGCM in the intermediate step. We now cited this publication and mentioned the absence of ARCM in his approach.

Main changes

Article:

Figure 2: new curves (t2m IPSLCM5A).

Section 3.3: new descriptions of the new figure 3.

Figure 3:    Annual mean precipitation, a) meridionally averaged (30 to 50°N), b) zonally averaged (-10 to 35°E), in the historical simulations.

Table 1: (former table 2) new column with the Black Sea values and the budget.

Figure 7: fix the contour/shading issues.

Figure 9: remove the difference (EHOL vs PICTRL)

Section 4.4: new description of the new figure 11

Figure 11: move the monthly Nile climatology to the supplements

Section 4.5:  move the first paragraph of the conclusion to 4.5

SOM:

Addition of figure 1: climatological runoff of the Nile River

Figure S2: addition of IPSLCM5 SST (raw and corrected)

Figure S8: Interannual evolution of the sea surface salinity (EHOL and PICTRL)

Table S1: (former figure S1)

**Reply to Anonymous Referee #2**

[Reviewer comment]
Review of GMD-2019-196 Vadseria et al. present a sequential modelling tool to investigate (paleo-)climate change effects on Mediterranean Sea circulation. They first explain their set-up and validate for the present-day. Then an example of application, the Early Holocene, is given. It seems like a valid approach that is indeed of use for multiple (paleo-) applications. I would however suggest revision to make the paper clearer, both structurally and with respect to what exactly the added value of their sequential modelling tool is.

[Reply]
We thank the reviewer for his/her constructive comments that help to improve our work. We have implemented all of them in the revised manuscript.

[Reviewer comment]
So my main comments are:
- structurally the paper can improve to clear up some unclarities. For instance, Fig. 2 states "hist-obs" while the text only mentions "hist". I guess you mean the same simulation. Also, many citations seem to be absent from the reference list.

[Reply]
We apologize for such confusions. We made the necessary correction accordingly.

[Reviewer comment]
- content-wise, the authors seem to claim that high-resolution atmospheric forcing is needed to get correct Mediterranean Sea circulation. This needs to be better substantiated by results or discussion. For instance, can you show that your simulation yields better results than, say, a OGCM run forced directly with AGCM forcing rather than ARGCM?

[Reply]
We try to demonstrate this point by using results from literature. Lebeaupin Brossier et al. (2011) showed that high-resolution atmospheric forcing was crucial in triggering the Mediterranean deep-water formation. Increasing the spatial resolution produces finer and more intense wind stress over the north western Mediterranean area. It also slightly modified the precipitating systems representation. Li et al (2006) also showed that the 50-km resolution in the atmosphere seems a threshold to induce the right Mediterranean overturning circulation.

[Reviewer comment]
Please find more detailed comments below, followed by the GMD review criteria.
P2, line 67 "the localization of the ... of debate": true, and actually your set-up would allow for testing separate forcing sources for sapropel formation (i.e. only adding additional freshwater to a certain location, or only precipitation versus only river runoff). This would make your model setup even more useful than using it for overall Med-Sea circulation under paleo-climate-forcings.

[Reply]

(All the lines mentioned hereafter refer to new version of the manuscript)

We agree that we may perform a series of sensitivity experiments to test the response of the Mediterranean overturning circulation to different forcings. Actually, we are working on the impact of different hydrological perturbations during the deglaciation on the Mediterranean oceanic dynamics. We hope to be able to present these new results soon. However, we want to keep our initial objective for this manuscript, to build a coherent modelling chain, able to go to detailed regional oceanic features from simulations with coarser-resolution global models.

[Reviewer comment]
P3 lines 73-77. Please provide section numbers when outlining the paper.

[Reply]
Done

[Reviewer comment]
P4 lines 130-140: how about the exchange with the Black Sea? Is it common to deal with as if a river?

[Reply]
Yes, in most Mediterranean modelling studies, when the Black Sea is not explicitly simulated, it is often treated as a river. It is actually the case for all studies using the NEMO-MED platform.

[Reviewer comment]
P5 section 1.3: in my opinion this fits better in the methods section, where it can be merged with the specific LMDZ-NEMO set-up.

[Reply]
Yes, that's right. The current structure of the manuscript reflects our intellectual confrontation between generality and particularity. Our philosophy was to firstly propose a general concept, and then fill up different boxes by nominative models. So, we want to keep that structure

[Reviewer comment]
P6 lines 188-190: mention where it can derive boundary conditions from (SIC and SST).

[Reply]
Boundary conditions (in particular, SST and SIC) are derived from global coupled models, from IPSL-CM5A in our actual implementation. We detailed this description in the revised manuscript.

[Reviewer comment]
P6 lines 199-200: give a reference for ORCHIDEE and is it run at the same resolution?

[Reply]
Yes, ORCHIDEE (the land surface model) was integrated into LMDZ. The two components work at the same resolution (, reference added l208).

[Reviewer comment]
P6 line 208: which 'first dataset of river discharges' do you refer to? And does this represent the majority of discharge in the 192 ORCHIDEE river mouths?

[Reply]
We apologize for the confusion. We modified the manuscript accordingly (l216). In fact, we had the choice to use a dataset of climatological river discharges. This dataset divided the Mediterranean draining basin into 33 river mouths. However, when the ORCHIDEE model is interactively used to calculate river discharges, there are 192 river mouths. The two approaches are independent, to be actually used by optional choice.

[Reviewer comment]
P7 lines 211-213: how realistic is the assumption that water from the Black Sea is fresh? And does the Q+P-E budget over the Black Sea derive from the AGCM orARCM?

[Reply]
It is a commonly-used treatment when the Mediterranean model doesn't include the Black Sea. The fresh water assumption is entirely justified although the actual water flow from the Black Sea can be salty, since what we evaluated in terms of E, P and Runoff is indeed the fresh water budget. What is important in the model is not the water mass itself, but the salt content. We made some revisions in the new manuscript for this regard.

[Reviewer comment]
P7 line 215 / fig 1: to fit the figure with all your simulations, can you include that boundary conditions can also derive from reanalysis?

[Reply]
It is theoretically possible to include boundary conditions deduced from re-analysis. But our main goal in the platform is to use global coupled simulations as a departure to conduct the whole chain.

[Reviewer comment]
P7 line 229: maybe put the table that shows an overview of experiments in the main text.

[Reply]
We prefer to let that table describing simulation parameters in the Supplementary materials, in order to put the modelling chain and the general concept into a more prominent position.

[Reviewer comment]
P7 line 239: the cited paper is not in the reference list (as are many other citations)

[Reply]
We apologize for this issue. We now double-checked the revised manuscript.

[Reviewer comment]
P8 line 246: "for one period" rather than "for a period"

P8 Fig 1: usually u is zonal wind, v is meridional wind.
P8 line 266: write out WOA

[Reply]
We corrected the manuscript accordingly.

[Reviewer comment]
P9 Fig 2: the legend mentions "HIST-OBS", I guess you mean experiment "HIST"?
Also, why do you use ERA20C here whereas experiment "HIST" is forced with ERA-Interim?

[Reply]
Yes, we corrected the legend and the caption of the graphic. We did not use ERA-interim, since it starts from 1979 only. ERA20C starts from 1970 and is more suitable for our purpose. During the revision, we re-drew the graphic, and improved the description on how different curves were calculated. We also added the global T2m from the ocean-atmosphere coupled model IPSL-CM5A, considered as a baseline, in order to appreciate the improvement that we have in our system.

[Figure]

**New Figure 2: Time series of annual mean surface air temperatures at 2 m in HIST (red) and ERA20C (black, ref: Stickler et al., 2014) and IPSLCM5A (green) for global average (solid lines) and Mediterranean-region (ocean and continent) average (dashed lines).**

[Reviewer comment]
P10 line 291: Table 2, not 3

[Reply]
Yes, we corrected that. In the revised manuscript, it becomes Table 1 summarizing all components of the fresh water budget.

[Reviewer comment]

P10 Fig 3: again a different dataset is used (CRU), whereas Fig. 2 compares to ERA20C, and "HIST" is forced with ERA-Interim. Why would you use such a range of datasets? And why not use a reanalysis that has values over the sea? Also, looking at the color scales, it seems that the overestimation is as large as the modelled precipitation itself over land. So the relative overestimation there is near 100%?

[Reply]
Thank you for your careful reading. We finally decided to change this plot to curves showing zonal and meridional averages. We also modified the relevant text accordingly in section 3.3.

[Figure]

**New Figure 3: Annual mean precipitation, a) meridionally averaged (30 to 50°N), b) zonally averaged (-10 to 35°E), in the historical simulations with AGCM (LMDZ-global) and ARCM (LMDZ-regional). Observation comes from GPCP (Global Precipitation Climatology Project, 1979 to 1999, blue line, ref: Adler et al., 2018). and ERA20C (green line, ref: Stickler et al., 2014).**

[Reviewer comment]
P11 Fig 5: in the upper panel it seems like there is a contour overlaying the colours, are those from observations?

[Reply]
No, they are not from observations. Contours in the upper panel are the maximum of MLD (mixed-layer depth) throughout the entire simulation.

[Reviewer comment]
P12 Table 1: provide units and define IS.
P12 line 337: refers to 5b, instead of 5a?
P12 line 340: Figure 6a instead of 7a.

[Reply]
Corrected

[Reviewer comment]
P13 lines 350-352: if there is still a lack of modelling capacity to simulate Med-Sea deep circulation, how can you verify that your study is an improvement?

[Reply]
We now removed this phrase which is not very relevant for our manuscript. There are some uncertainties concerning the changes in deep circulation for the Mediterranean Sea. Our simulation is nevertheless in the range of circulation changes provided by different modeling studies. This is encouraging for us to go a step further and to investigate a larger perturbation, such as the early Holocene. In that context, we added some new text in the revised manuscript (l410):
"The simulation of the thermohaline circulation is well captured by the oceanic model and in the range of the state of the art of existing Mediterranean regional models (compared to the simulations of Adloff et al., 2015 and Somot et al., 2006 for instance). This feature inspires confidence in our modelling platform for the investigations of past climate."

[Reviewer comment]
P14 lines 362-364: Figures 2 and 4 show that your simulation results in significantly lower temperatures than observed, yet here you say they are consistent?

[Reply]
Yes, there are cold biases. We changed the corresponding text in the revised manuscript "The atmospheric simulation is acceptable compared with observations for the air temperature at 2m at both global and regional scales "(l405).

[Reviewer comment]
P14 line 365: How can a model overestimate the precipitation over the surrounding land substantially (fig 3) yet have precipitation over the sea close to observation (Table 2) and have lower river runoff than HIST or PICTL (with overestimation of precipitation over land, why is runoff not overestimated too – is this due to bias correction?)

[Reply]
Yes, it is possible for a model to have roughly right precipitation over the Sea, but too much precipitation over the surrounding land. Our model did show such a feature for its basic climatology, and for changes from PICTRL to EHOL. We now calculated all components of the fresh water budget, and discussed their variation among the three simulations (HIST, PICTRL and EHOL, section 3.3 and 4.4). Rivers discharges increase significantly from PICTL to EHOL, making the fresh water deficit to decrease.

[Reviewer comment]
P15 section 3.2: is there any additional ice sheet remaining in the early Holocene in the model?

[Reply]
No, no more remaining ice sheets for the early Holocene, in our model at least. We used a simulation in equilibrium for 9ka using the orbital forcing appropriate for this period with no more Fennoscandian and Laurentide ice sheet (FIC, LIS). Therefore, the sea level also remains unchanged (as to present day)

[Reviewer comment]
P16 line 398-399: "increased Early Holocene summer insolation" or "increased Early Holocene insolation seasonality".
P16 line 400-404: refer to figures 7c, 7d.
P17 Figure 7: in the caption the "a" after "b) summer temperatures" should be removed

[Reply]
Corrected

[Reviewer comment]
P20 line 494-497: how does the increased Nile runoff in PICTRL (do you mean compared to observations?) compare to the overall lower runoff reported in table 2?

[Reply]
As for HIST, the river runoff for PICTRL is not calculated with the precipitation of the model. PICTRL river runoff is the same as HIST (so prescribed) but with Pre-damming Nile value.

[Reviewer comment]
P21 Fig 11: Especially in late winter and summer, runoff from the Black Sea is decreased by roughly the same order of magnitude as the increase in Nile runoff. Can you reflect on the possible role that the Black Sea runoff alone could have in sapropel formation?

[Reply]
For summer the runoff decrease of the Black Sea is quite "marginal" compared to the Nile increase (-6000/+45000 m3/s). Actually the role of the Black Sea during the Early Holocene is overall quite marginal but some studies pointed out that a freshwater release was likely throughout the deglaciation (as Chepalyga, 2007, Soulet et al., 2011, 2013) , due to the Fennoscandian Ice sheet melting, and thus affect the Aegean Sea and maybe the Eastern Basin during this period.

[Reviewer comment]
P22, lines 522-525: what do you mean by the reference for correction is the preindustrial state? How is river runoff corrected based on pre-industrial climate?

[Reply]
We choose to "correct" the Mediterranean river runoff during the Early Holocene based on the precipitation difference (EHOL – PICTRL) coming from both the ARCM and AGCM and apply it to the PICTRL river runoff (which was prescribed). The procedure of river runoff is detailed in the supplementary material (**Text S2: Bias correction**)

[Reviewer comment]
P22 lines 543-545: I would not say that your simulations show similar changes as Adloff or Bosmans. For instance Adloff (their fig 9) shows strong salinity increases around Greece, and Bosmans (their fig 11) do not show a decreased mixed layer depth in the Ionian sea.

[Reply]

The reviewer is right, we modify the text l598: "Our oceanic simulation depicts these behaviours well and is overall similar"

[Reviewer comment]
P24 Fig 13: add to caption that this can be compared to Fig 6 (PICTRL).

 [Reply]
Done

[Reviewer comment]
P24 line 562: "for the first time" – this you could mention more clearly in the introduction.

[Reply]
Thanks. We added a new phrase for this regard: "To tackle this issue, a sequential architecture of a global-regional modelling platform has been developed for the first time and is described in detail in this paper" (l22).

[Reviewer comment]
P25 lines 571-579: this is not a section that should be in the Conclusions. It is more fitting for a discussion section. It also makes me wonder if there is anything known of the effect of keeping the Bosphorus exchange as it is today.

[Reply]
It is not easy to conclude on the role of the Bosphorus during the S1 period. According to the review and the synthetic work of Rohling et al., 2015, it is quite established that the Black Sea, through the Bosphorus, was not a major freshwater source during S1, so that is why we remain that parameter as it was set in HIST and PICTRL. As the reviewer suggested we moved this paragraph at the end of the 4.5 section

[Reviewer comment]
P26 References: make sure all cited literature is in the reference list.

[Reply]
We apologize for this issue concerning the cited references. We double-checked it during the revision.

Main changes

Article:

Figure 2: new curves (t2m IPSLCM5A).

Section 3.3: new descriptions of the new figure 3.

Figure 3:  Annual mean precipitation, a) meridionally averaged (30 to 50°N), b) zonally averaged (-10 to 35°E), in the historical simulations.

Table 1: (former table 2) new column with the Black Sea values and the budget.

Figure 7: fix the contour/shading issues.

Figure 9: remove the difference (EHOL vs PICTRL)

Section 4.4: new description of the new figure 11

Figure 11: move the monthly Nile climatology to the supplements

Section 4.5: move the first paragraph of the conclusion to 4.5

SOM:

Addition of figure 1: climatological runoff of the Nile River

Figure S2: addition of IPSLCM5 SST (raw and corrected)

Figure S8: Interannual evolution of the sea surface salinity (EHOL and PICTRL)

Table S1: (former figure S1)

[revised manuscript text omitted]
 compared to present-day observations (Figure S1). The bias-corrected  we applied is based on the observed climatological runoff (Ludwig et al. 2009; Vorosmarty et al., 1998) and the differences between the Early Holocene simulation and present-day simulation. When the difference is relatively not significant, the corrected runoff is set to the climatology, mainly to avoid negative values[1]. However, in order to stay consistent with the methodology for SST and SIC bias correction, we chose the absolute difference correction method for the river runoff.

**Text S3: Comparison of model simulation outputs and reconstructed data for the Mediterranean basin**

*Continental precipitation*

The reconstructed data used for  comparison with the EHOL simulation is taken from (Dormoy et al. (2009) for the Aegean Sea, from (Peyron et al. (2011) for the Lake Accesa and  Tenaghi

Philippon, and (Magny et al. (2013) for Lake Pergusa. In these studies, continental precipitation is
* * *
[1] Namely, when the difference does not exceed 25%, of the annually average annual difference for the Nile river runoff (due to the simulated amplitude, cf section 4.4) and 5% for the rest of the rivers.

reconstructed based on pollen sequences to emphasis the changes in precipitation seasonality. Several methods are used to determine these changes. We chose to reconstruct these changes using the Modern Analogue Technique (MAT, Guiot, 1990), because, in their study, Magny et al. (2013) compared their data to Peyron et al. (2011). We extracted data values framing a few hundred years around 9.5 ka cal BP, consistent with the orbital parameters of our atmospheric simulations (both global and regional). For the Northern Sahara, data are based on $\delta^{18}O$ from Bar-Matthews et al.( 2003).

Comparison between model outputs and reconstruction data in terms of annual and seasonality changes can be conducted and anomalies against modern values can be shown. In winter, the model shows positive precipitation anomalies for the four sites (Lake Accesa, model: +20-36 mm, data: +20-40mm, Tenaghi Philippon, model: +30-45 mm, data: +10-35 mm, Aegean, model: +29-45 mm, data: +10-80mm, Lake Pergusa, model: +7-26 mm, data: +35-60mm, Table S1). In summer, the model shows a more contrasted response, with negative anomalies in summer temperatures (1, Table S1) due to the homogenous drought (ig 10d in the main article). However, this comparison cannot reflect the precipitation changes for the entire continent. Indeed, in north of Lake Accesa we see positive summer anomalies (ig 10d in the main article).

*Sea Surface Temperatures*

We conducted a comparison of model output and data for SST as Adloff et al., (2011) did with the reconstruction of Kucera et al., (2011) (unpublished work). This reconstruction is based on census counts of foraminiferal species, and on the artificial neural network for the transfer function. The data used span the Holocene Insolation maximum interval (8.5 - 9.5 ka BP). Winter SST values (January to March, igure S22, f) are a bit lower than the reconstruction especially for the Eastern basin (-1 to -2 °C). The simulated summer SSTs (July to September, igure S3, j) are higher between the Tyrrhenian Sea and the Levantine Sea (+1 to +4 °C). This enhanced contrast between winter and summer values for simulated SST produced an annual signal in good agreement with the reconstructed values (igure S3, c). Our results depict the same signal pattern as the simulations of Adloff et al. (2011) do, with some differences in the enhanced seasonal contrast. In igure S2 are also depicted the same climatology for the bias Early Holocene SST and the bias corrected Early Holocene SST boundary conditions used in the model architecture. This figure shows how the SST signal have been improved, from the bias correction to the ORCM simulation, in order to range the reconstruction with the use of the regional models.

*Sea Surface Salinities*

The comparison of SSS over the Mediterranean Sea provides an appropriate indicator of freshwater perturbation induced by enhanced river flux. As a reference for comparison, we use a synthesis  of SSS  sampled from the S1 deposition, and provided by Kallel et al. (1997). Our EHOL simulation takes the Nile river enhancement into account, that is an annual river discharge of 13000 $m^3$.s$^{-1}$, against 2930 $m^3.s^{-1}$  for the pre-industrial value), The North-East river (Buyukmenderes, Vardar, Acheloos, Vjosa, Semanit, Shkumbin, Durres, Mat and Drini) have their annual fresh water discharges increasing from 1082 $m^3.s^{-1}$  at pre-industrial level to  1622 $m^3.s^{-1}$ . The fresh water discharge from February to May increases even more, from 1619 $m^3.s^{-1}$  at pre-industrial level to  3228 $m^3.s^{-1}$  for EHOL. Our EHOL simulation, even with a significant increase of freshwater input, still cannot reproduce a sufficient decrease of SSS  to match the reconstructed values, as shown in Figure S4.  .  Rohling (1999, 2000) pointed out that this mismatch can be partly attributed to uncertainties in salinity reconstruction. It is not always straightforward to interpret the isotopic composition of oxygen in terms of salinity. Finally, it is likely that an additional non-negligible fresh water source is missing. To explain the substantial SSS decrease, an additional source of freshwater associated with an amplification of the flux of the North African rivers could potentially be superimposed on the Nile. Indeed, changes of this type in the hydrology are clearly indicated by the data but are not reproduced in most of the Early and Mid-Holocene simulations.

[Figure]

**Figure S1: climatological runoff of the Nile River, observed pre-damming values (red), runoff as**

**simulated by the ARCM, PICTRL (blue) and EHOL (green), and corrected runoff used in the**

**ORCM.**

[Figure]

**Figure S1: Model-data comparison for continental precipitation (solid lines = EHOL simulation,**
**dashed lines = pollen data reconstruction). First row: Lake Accesa (Northern Italy) (Peyron et al.,**
**2011), Second row: Tenaghi Philippon, (Greece) (Peyron et al., 2011), Third row: Lake Pergusa**
**(Sicily), (Magny et al., 2013), Fourth row: Aegean Sea, (Dormoy et al., 2009), Fifth row: Northern**
**Sahara (Bar-Matthews et al., 2003). First column: winter precipitation, Second column: summer**
**precipitation, Third column: annual precipitation.**

[Figure]

**Figure S42: Model-data comparison for SST, adapted from Adloff (2011). Dots represent the unpublished synthesis of Kucera et al. (2011), published in Adloff (2011). The background colour represents, in the first column, the bias SST boundary conditions (BC) derived from the Early Holocene IPSL-CM5 simulation (AMIP resolution), in the second column, the bias corrected SST BC as it has been used to drive the AGCM and the AGCM both (AMIP resolution), and, in the third column, SST in the EHOL experiment realized with the ORCM (1/8°, averaged over the last 30 years of simulation). **

[Figure]

**Figure S53: Model-data comparison for SSS. Dots represent the synthesis of Kallel et al. (1997a). The background colour represents the EHOL simulation.**

[Figure]

**Figure S44: Interannual evolution of the index of stratification (IS) for the Mediterranean Sea**

**for the HIST simulation (including the spin-up phase).**

[Figure]

**Figure S55: Interannual evolution of the Zonal overturning Stream Function (ZOF) in the**

**eastern Mediterranean Sea for the HIST simulation (including the spin-up phase).**

[Figure]

**Figure S86: Interannual evolution of the index of stratification (IS) for the Mediterranean Sea for the PICTRL and EHOL simulations (including the  PICTRL spin-up phase).**

[Figure]

**Figure S97: Interannual evolution of the Zonal overturning Stream Function (ZOF) in the eastern Mediterranean Sea for the PICTRL and EHOL simulations (including the PICTRL spin-up phase).**

[Figure]

**Figure S8: Interannual evolution of the sea surface salinity (SSS) for the Mediterranean Sea for**

**the PICTRL and EHOL simulations (including the PCRL spin-up phase).**

| Precipitation (mm) | Winter | | | Summer | | | Annual | | |
|---|---|---|---|---|---|---|---|---|---|
| | MODERN | ΔOBS | ΔEHOL | MODERN | ΔOBS | ΔEHOL | MODERN | ΔOBS | ΔEHOL |
| Lake Acessa | 240 | 20-40 | 20-36 | 80 | 0-30 | (-26)-(-8) | 750 | 10-70 | 8-60 |
| Tenaghi Philippon | 225 | 10-35 | 30-45 | 80 | 20-50 | (-13)-5 | 600 | 130-225 | 17-49 |
| Lake Pergusa | 225 | 35-60 | 7-26 | 80 | 30-50 | (-17)-(-3) | | | |
| Aegean Sea | 200 | 10-80 | 29-45 | 40 | 0-40 | (-19)-0 | | | |
| Northern Sahara | | | | | | | <200 | 700-800 | (-20)-15 |

**Table S1: Model-data comparison for continental. First row: Lake Accesa (Northern Italy) (Peyron et al., 2011), Second row: Tenaghi Philippon, (Greece) (Peyron et al., 2011), Third row: Lake Pergusa (Sicily), (Magny et al., 2013), Fourth row: Aegean Sea, (Dormoy et al., 2009), Fifth row: Northern Sahara (Bar-Matthews et al., 2003). "MODERN" refers to the present values of precipitation, "OBS" to the data (around 9.5 ka cal BP), and "EHOL" for the Early Holocene simulation described in the article.**

| | HIST | PICTRL | EHOL |
|---|---|---|---|
| Orbital parameters | e = 0.01672
ε = 23.44
ω -180 = 102.7 | IdemSame as in HIST | e = 0.01935
ε = 24.231
ω -180 = 303.3 |
| Atmospheric $CO_2$ | Annual observed global mean (1970-1999) | 2840 ppm | 264884 0 ppm |
| SST forcing | Era-Interim monthly forcing (1970-1999 | IPSL-CM5A picontrol + SST correction | IPSL-CM5A early Holocene + SST correction |
| SIC forcing | Era-Interim monthly forcing (1970-1999 | IPSL-CM5A picontrol + SIC correction | IPSL-CM5A Early Holocene + SIC correction |

**Table S21: Forcings and parameters used in both AGCM and ARCM. ε is the elliptic orbit obliquity, e, the eccentricity and ω, the longitude of the perihelion. The reader can notice that the**

**pCO$_2$ should be 260 ppm as suggested by the PMIP protocol for mid-Holocene. The goal in this**
**paper was mainly to have a sensitivity to orbital parameters.**

[revised manuscript text omitted]

**Requested minor revisions to gmd-2019-196**

Dear Lauren Gregoire,
We are very pleased about your decision about the manuscript.
We also thank you for the new comments you addressed. Please see our reply.

L228: Clarify the procedure for boundary conditions applied when running NEMO ocean only model. This paragraph is currently too vague and It's difficult to make sense of it. Clarify what the flux are, what the restoring term is and what the constant coefficient controls. If there is a paper you can cite that would be great, otherwise please include an equation.

Indeed, that the flux and restoring term comes from Barnier et al., 1995, we inserted the reference in the new version of the manuscript (L226 and L715).

L306: "The global simulation, after SST bias correction, ranged with the observation, compared to IPSLCM5A (Figure 2)." "ranged with the observation" is not clear. Do you mean "has the same range of variability as e.g. the 2m temperature over the Mediterranean region from ERA20C" ?

We apologize that there was a misunderstanding in our results description. Actually, the line you mentioned only describes the global average of 2m temperature and it is not related to the Mediterranean. We suspect that the word "range" was inappropriately used. We made the necessary changes in the new manuscript. By the way, we also make a slight change for the regional aspect over the Mediterranean:

L302

Previous sentence: "The global simulation, after SST bias correction, ranged with the observation, compared to IPSLCM5A (Figure 2)"

New sentence: "The global simulation (continued red curve in Fig. 2), after SST bias correction, is very close to the observation (continued black curve), with a tremendous improvement compared to IPSLCM5A (green curve in Figure 2)." The regional model reproduces the warming trend and aspects of the interannual variability close to observations, **but with a mean cold shift of about -0.6°C.**

L219 "A first dataset of climatological river discharges is proposed by default to cover the entire Mediterranean draining basin with represents 33 river mouths." What do you mean by "is proposed" to whom, for what? How is it used? Please clarify the text.

We apologize again for the confusion. Actually, that dataset of rivers freshwater discharges was constructed by the initial NEMOMED model developers, and was proposed to us (authors of the manuscript) to be used if we don't include the rivers interactively. We changed this paragraph into past tense ("is" to "was", L216), which can help to remove the confusion. "A dataset of climatological river discharges was proposed by default within the NEMOMED8 platform to cover the entire Mediterranean draining basin with 33 river mouths."

"[Reviewer comment]

P7 lines 211-213: how realistic is the assumption that water from the Black Sea is fresh? And does the Q+P-E budget over the Black Sea derive from the AGCM orARCM?
[Reply]
It is a commonly-used treatment when the Mediterranean model doesn't include the Black Sea. The fresh water assumption is entirely justified although the actual water flow from the Black Sea can be salty, since what we evaluated in terms of E, P and Runoff is indeed the fresh water budget. What is important in the model is not the water mass itself, but the salt content. We made some revisions in the new manuscript for this regard."
I can't see how and where you have addressed this point. Please clarify with citation of the text.

For this purpose, we update the text at l221 "The Black Sea fresh water assumption comes from NEMOMED modeling community that consider it as a yearly source of freshwater" and l450 "For this part, the water budget over the Black Sea is calculated from the ARCM output"

"[Reviewer comment]
P14 lines 362-364: Figures 2 and 4 show that your simulation results in significantly lower temperatures than observed, yet here you say they are consistent?
[Reply]
Yes, there are cold biases. We changed the corresponding text in the revised manuscript "The atmospheric simulation is acceptable compared with observations for the air temperature at 2m at both global and regional scales "(l405)."
I am not satisfied with how you have addressed this point. Changing the word "consistent" with "acceptable" just makes a wrong statement into something vague and subjective. Please quantify and describe the cold bias and discuss the implications here.

Indeed, we made the following correction l417: "Validation of our platform was based on the historical period from1970 to 1999. after bias correction of global SST, the 2-m surface air temperature in the HIST global simulation is comparable to the observational counterpart. However, the simulated surface air temperature within the regional model is colder (as shown in Figure2), which implies SST cold biases for the Mediterranean Sea"

"[Reviewer comment]
P22, lines 522-525: what do you mean by the reference for correction is the preindustrial state? How is river runoff corrected based on pre-industrial climate?
[Reply]
We choose to "correct" the Mediterranean river runoff during the Early Holocene based on the precipitation difference (EHOL – PICTRL) coming from both the ARCM and AGCM and apply it to the PICTRL river runoff (which was prescribed). The procedure of river runoff is detailed in the supplementary material (**Text S2: Bias correction**)"
This point hasn't been addressed adequately, please correct the text when you make reference to the "correction" to clarify that you apply a bias correct as described in Text S2.

If we understand well, we just need to clarify the response in the manuscript l593. « The procedure of river runoff is detailed in the supplementary material (Text S2: Bias correction) »

Editorial corrections (suggested changes in bold):

Thanks, we took all of these suggestions into account.

L 153: "This architecture is based on a method **that provides** as much **compatibility** as possible amongs**t** the models used and high **consistency** with data."

L224L "river mouths **that** cover the …"

L330 " both the precipitation and evaporation over the Mediterranean Sea in HIST **are** very close to the observations" Quantify how close.

L331"but both the precipitation and evaporation over the Mediterranean Sea in HIST are very close to the observations**, with 10 mm.yr$^{-1}$ of oceanic precipitation difference between HIST and the mean observation value (by taking the upper and the lower value), and 18 mm.yr$^{-1}$ for the oceanic evaporation.**"

L331 "**The** two other simulations **included in Table 1**, PICTRL and EHOL, are those designed to investigate**s** the Early Holocene climate **(see Section 4)**."

L407: "The ZOF **in HIST depicted in Figure 6)**"

Figure 6: replace row numbers with figure labels (a-h)

L425 It is not correct to say "ranges with the observation". Do you mean "has the same range of variability as the observations"

Please see the new version of the main manuscript, the supplement remains unchanged.

Regards

Tristan Vadsaria on Behalf of all co-authors

Added reference L715:

[revised manuscript text omitted]
 from1970 to 1999. after bias correction of global SST, the 2-m surface air temperature in the HIST global simulation is comparable to the observational counterpart. However, the simulated surface air temperature within the regional model is colder (as shown in Figure2), which implies SST cold biases for the Mediterranean Sea   The simulated precipitation from the atmospheric models produces a signal that has the same range of variability as the observations, but there is significant overestimation of precipitation over the mountainous area and over the land surrounding the Mediterranean Sea. However, the freshwater budget over the sea is close to observations for both evaporation and precipitation. The areas of intermediate and deep convection produced by the model are realistic, and the simulation of the thermohaline circulation is well captured by the oceanic model and in the range of the state-of-the-art existing Mediterranean regional models (compared to the simulations of Adloff et al., 2015 and Somot et al., 2006 for instance) and reanalysis as well (Pinardi et al., 2019). These features inspire confidence in our modelling platform for the investigations of past climate.

**4  Application of the modelling chain to the Early Holocene**

In this section, results obtained when our sequential modelling chain is applied in a paleoclimate context are presented, which was our initial motivation for developing this modelling tool. We chose to test the performance of our tool on the Early Holocene, a period marked by significant changes in climate and ocean dynamics over the Mediterranean basin, when the last sapropel event, S1, occurred in the Mediterranean Sea. Our experimental design relies on the comparison of two simulations: the Early Holocene (EHOL) with PICTRL based on pre-industrial conditions, the latter acting as a reference.

**4.1  Experimental design**

As indicated in the general flowchart of our modelling platform, global SST and SIC are required to initiate our sequential modelling. The basic assumption is that the climate change signal can be reconstructed from global SST and SIC, an accepted practice within the climate modelling community. In this study, two existing long-term coupled simulations from IPSL-CM5A is used, one covering the pre-industrial period and the other covering the Early Holocene (around 9.5 ka). Taking the last 100 years of each simulation, a climatological SST and SIC is constructed. After conducting bias-correction, these outputs from IPSL-CM5A are then used to drive the AGCM (LMDZ-global) and the ARCM (LMDZ-regional) in a further step. The duration of the PICTRL and EHOL atmospheric simulations is 30 years (both global and regional models).

Oceanic temperature and salinity in the Atlantic buffer-zone, as well as freshwater discharges from Mediterranean rivers, are all bias-corrected for NEMOMED8, as described in the general methodology. However, it needs to be pointed out that the reference point for the Nile river discharge is not modern observations but is set at pre-industrial values ($2930 \ m^3.s^{-1}$ for annual mean, Vorosmarty et al., 1998)

corresponding to a period before construction of the Aswan dam. For this part, the water budget over

[revised manuscript text omitted]